# Cytosolic TOP3α facilitates mitochondrial DNA sensing by cGAS

Dongjing Cai[1,2,7], Cheng Chen[2,3,7], Piyanat Meekrathok [1,2], Weiqian Zeng[1,2], Zheng Wang [2,3], Zhigang Peng[1,2], Yunan Mo[1,2], Xia Xu[2,4], Junling Wang[2,3] & Jian Qiu [1,2,5,6]✉

## Abstract

Mitochondrial DNA (mtDNA) serves as a potent activator for cellular innate immune responses. Topoisomerase 3α (TOP3α), a type IA topoisomerase, is canonically localized to mitochondria and nuclei, but its enigmatic cytosolic fraction—observed over two decades ago—has remained functionally undefined. Here, we uncover a critical role for cytosolic TOP3α in amplifying mtDNA-triggered innate immunity. We observe that aberrant TOP3α expression causes mtDNA clustering and release via mPTP-VDAC, stimulating cGAS-mediated inflammatory responses. Cytosolic TOP3α facilitates the sensing of released mtDNA by cGAS and amplifies downstream innate immune signaling. Using an in vitro cell-free system, we reveal that TOP3α directly augments mtDNA interaction with cGAS, which in turn competes with TOP3α for mtDNA binding. A rare mutation of a highly conserved residue (G250D) of TOP3α impairs the assembly of TOP3α polypeptides into protein complexes and its binding to mtDNA. Furthermore, mutant TOP3α hinders cGAS-mtDNA interaction and compromises cGAS-driven immunity. Our findings reveal a function for cytosolic TOP3α as a regulator for cGAS-driven inflammation.

**Keywords** Mitochondrial DNA; Cytosolic TOP3α; cGAS; Inflammation
**Subject Category** Immunology

## Introduction

Mitochondria are essential organelles for various cellular activities including ATP production, reactive oxygen species (ROS) generation and innate immune signaling (Nunnari and Suomalainen, 2012). A mitochondrion contains its own DNA genome which is maintained and expressed under the intricate control of hundreds of nuclear encoded proteins (Gustafsson et al, 2016). In mammals, mitochondrial DNA (mtDNA) is circular double-stranded of approximately 16.5 kilobase pairs, encoding 13 proteins crucial

for oxidative phosphorylation (OXPHOS) (Gustafsson et al, 2016). Defects in mtDNA replication and transcription caused by genetic perturbation or oxidative stress promote the generation of immunostimulatory nucleic acid species which can be released into the cytosol to stimulate inflammatory responses (West et al, 2015; Xian et al, 2022). As such, the integrities of mitochondrial inner and outer membranes modulated by membrane permeability transition pore (mPTP) and VDAC oligomerization serve as important barriers to restrain the engagement of cytosolic DNA sensors, such as cGAS, with mtDNA derived fragments (Andreeva et al, 2017; Civril et al, 2013; García and Chávez, 2007; Kim et al, 2019).

The contour length of mammalian mtDNA is about 5 μm, which needs to be compacted into mitochondria with a typical width of only 0.5 μm (Nass, 1966). Mitochondrial transcription factor A (TFAM) is an important mtDNA-binding protein required for the maintenance and expression of mitochondrial genome (Ngo et al, 2011; Shi et al, 2012). The cross-strand binding of TFAM molecules to mtDNA is crucial for compacting the lengthy mtDNA into a nucleoid with a diameter around 100 nm (Kukat et al, 2015; Kukat et al, 2011). Insufficient amount of TFAM causes aberrant mtDNA packaging and instability, promotes the release of mtDNA fragments into the cytosol, and triggers inflammatory responses via cGAS/STING pathway (Kukat et al, 2015; West et al, 2015). In the cytosol, released TFAM could prearrange mtDNA fragments into U-turn structures which facilitate cGAS dimerization and activation (Andreeva et al, 2017; Ngo et al, 2011). On the other hand, TFAM is also reported as an autophagy receptor when released into the cytosol to facilitate the autophagic clearance of released mtDNA thus limiting inflammation (Liu et al, 2024).

Topoisomerase 3α (TOP3α, encoded by *TOP3A*) belongs to the type IA topoisomerase subfamily and has a dual localization in both mitochondria and nucleus (Wang et al, 2002). The N-terminal 25 amino acids function as a mitochondrial targeting signal (MTS) to direct the cytosolic-translated polypeptide into the matrix of mitochondria. Upon reaching the matrix, TOP3α plays an important role in the decatenation and segregation of mtDNA once the replication is completed, facilitating the homeostasis of mitochondrial genome (Nicholls et al, 2018). In the nucleus, TOP3α cooperates with other proteins to resolve double Holliday

[1]Hunan Key Laboratory of Molecular Precision Medicine, Department of Neurosurgery, Xiangya Hospital, Central South University, Changsha 410008, China. [2]National Clinical Research Center for Geriatric Disorders, Xiangya Hospital, Central South University, Changsha 410008, China. [3]Department of Neurology, Xiangya Hospital, Central South University, Changsha 410008, China. [4]Department of General Practice, Xiangya Hospital, Central South University, Changsha 410008, China. [5]MOE Key Lab of Rare Pediatric Diseases & Hunan Key Laboratory of Medical Genetics & Hunan Key Laboratory of Animal Models for Human Diseases, School of Life Sciences, Central South University, Changsha 410008, China. [6]Furong Laboratory, Changsha, Hunan 410008, China. [7]These authors contributed equally: Dongjing Cai, Cheng Chen. ✉E-mail: qiujian@sklmg.edu.cn

junction during homologous recombination (Raynard et al, 2006). The pathogenic mutations of TOP3α (such as M100V, A176V and D479G) have been associated with numerous diseases (including mitochondrial diseases and Bloom syndrome) with a broad clinical spectrum (de Nonneville et al, 2022, Erdinc et al, 2023; Jiang, Jia et al, 2021; Martin et al, 2018; Primiano et al, 2022). Curiously, a minor fraction of TOP3α has also been observed to be present in the cytosol, but the functional significance is completely unknown (Wang et al, 2002). Since cGAS preferentially interacts with structured DNA ligands in the cytosol for innate immune signaling (Andreeva et al, 2017), it is intriguing to know whether cytosolic TOP3α could modulate DNA sensing by cGAS.

In this study, we report that aberrant expression of TOP3α caused mtDNA clustering and promoted the release of mtDNA via mPTP-VDAC axis to stimulate cGAS-mediated inflammatory responses. Cytosolic TOP3α directly facilitated the sensing of released mtDNA by cGAS, which in turn competed with TOP3α for mtDNA binding. A rare mutation of the highly conserved amino acid (G250D) in the noncatalytic CAP domain impaired the assembly of TOP3α protein complexes and mtDNA recognition. Furthermore, the mutant TOP3α diminished cGAS-mtDNA interaction and the downstream innate immune signaling. This study highlights the importance of cytosolic TOP3α on cGAS-mediated inflammatory responses.

## Results

### TOP3α downregulation causes mtDNA clustering and release

TOP3α is an essential protein required for decatenation and segregation of newly synthesized mtDNA from the original template after replication (Nicholls et al, 2018). To explore further possible effects of TOP3α depletion on mtDNA distribution and mitochondrial morphology, we downregulated TOP3α expression by two different small interfering RNAs (siRNAs) (Fig. 1A). The mitochondrial morphology and the membrane potential across mitochondrial inner membrane (assessed by OPA1 immunofluorescent staining and membrane potential sensitive dye MitoTracker Red) were not obviously affected by TOP3α depletion (Fig. EV1A; Appendix Fig. S1). Interestingly, the morphology of mitochondrial nucleoids visualized by DNA and TFAM staining was aberrant upon TOP3α downregulation (Fig. EV1B). Further super-resolution imaging revealed that depleting TOP3α reduced the number of mtDNA foci with concomitantly increased size (Fig. 1B–E), supporting the role of TOP3α in mtDNA segregation (Nicholls et al, 2018). Importantly, some of these aberrant mtDNA foci were not confined in the area labeled by mitochondrial outer membrane protein TOM40, indicating that TOP3α depletion promoted mtDNA clustering as well as the release of mtDNA from mitochondria (Figs. 1F–H and EV1C–G).

### Aberrant TOP3α expression stimulates cGAS-mediated inflammation via released mtDNA

To extract the released mtDNA in cytoplasm, we used digitonin-based method as previously described(Kim et al, 2019; Yu et al, 2020). Cellular components were extracted by different

concentration of digitonin and analyzed by western blotting. We observed that cytosolic proteins such as α-Tubulin were efficiently extracted at a concentration of digitonin as low as 0.025%, which did not extract mitochondrial or nuclear components (Fig. 2A; Appendix Fig. S2A). Subsequently, cytosolic components were extracted at this condition for quantitative PCR (qPCR) analysis of released mtDNA. As we observed in immunofluorescent analysis, knocking down TOP3α promoted mtDNA release in different types of cells derived from human or mouse (Fig. 2B,C,E). Moreover, elevated level of cytosolic mtDNA stimulated the expression of inflammatory genes (Fig. 2D,F). The release of mtDNA has been reported to be facilitated by the mitochondrial permeability transition pore (mPTP) of mitochondrial inner membrane and VDAC oligomerization of mitochondrial outer membrane (Kim et al, 2019; Xian et al, 2022). Indeed, inhibiting the mPTP by cyclosporin A (CsA) or VDAC oligomerization by VBIT-4 both impaired mtDNA release and inflammatory responses caused by TOP3α depletion (Fig. 2E,F). Therefore, depleting TOP3α promotes the fragmentation and release of mtDNA via mPTP-VDAC axis to stimulate inflammatory responses. Similar effects were observed when disturbing the compaction of mitochondrial nucleoids by depleting TFAM (Fig. EV2A–C). To ascertain whether the released mtDNA upon TOP3α depletion stimulates innate immune signaling via cGAS/STING pathway (Yu et al, 2020), we co-transfected cells with siRNAs against TOP3α and cGAS. The suppression of cGAS level significantly diminished the expression of inflammatory genes upon TOP3α depletion (Fig. 2G,H).

Intriguingly, mitochondrial TOP3α overexpression induced mtDNA aggregation (Fig. EV3A,B) morphologically resembling pathological mtDNA structures observed upon TOP3α or TFAM depletion (Figs. 1C and EV1B) (Nicholls et al, 2018; West et al, 2015). This finding aligns with established links between mtDNA stress and innate immune activation (Victorelli et al, 2023; West et al, 2015). Notably, TOP3α overexpression triggered mtDNA release and inflammatory responses without significantly affecting mitochondrial respiratory chain complexes, membrane potential and ROS level (Fig. EV3C–G). These data demonstrated that disturbing the proper expression level of TOP3α causes aberrant mtDNA aggregation and mtDNA release, thereby triggering inflammatory responses via cGAS.

### Cytosolic TOP3α promotes sensing of released mtDNA by cGAS

TOP3α has been reported to dually localize to both mitochondria and nucleus (Wang et al, 2002). Interestingly, a cytosolic pool of TOP3α was also observed, but was not functionally characterized (Wang et al, 2002). To unambiguously demonstrate the presence of cytosolic TOP3α, we homogenized cells treated with control or TOP3α siRNA, and separated cytosolic fraction from the rest of cells by differential centrifugation. A minor pool of TOP3α was detected in the cytosolic fraction, and was specifically depleted by siTOP3α treatment (Fig. 3A). Importantly, neither nuclear nor mitochondrial proteins were detected in the cytosolic fraction, but cytosolic protein α-Tubulin was clearly present (Fig. 3A; Appendix Fig. S2B).

To analyze how cytosolic TOP3α may impact the interaction of released mtDNA with cGAS, cytosolic components were extracted and incubated with biotin-labeled mtDNA fragments. After

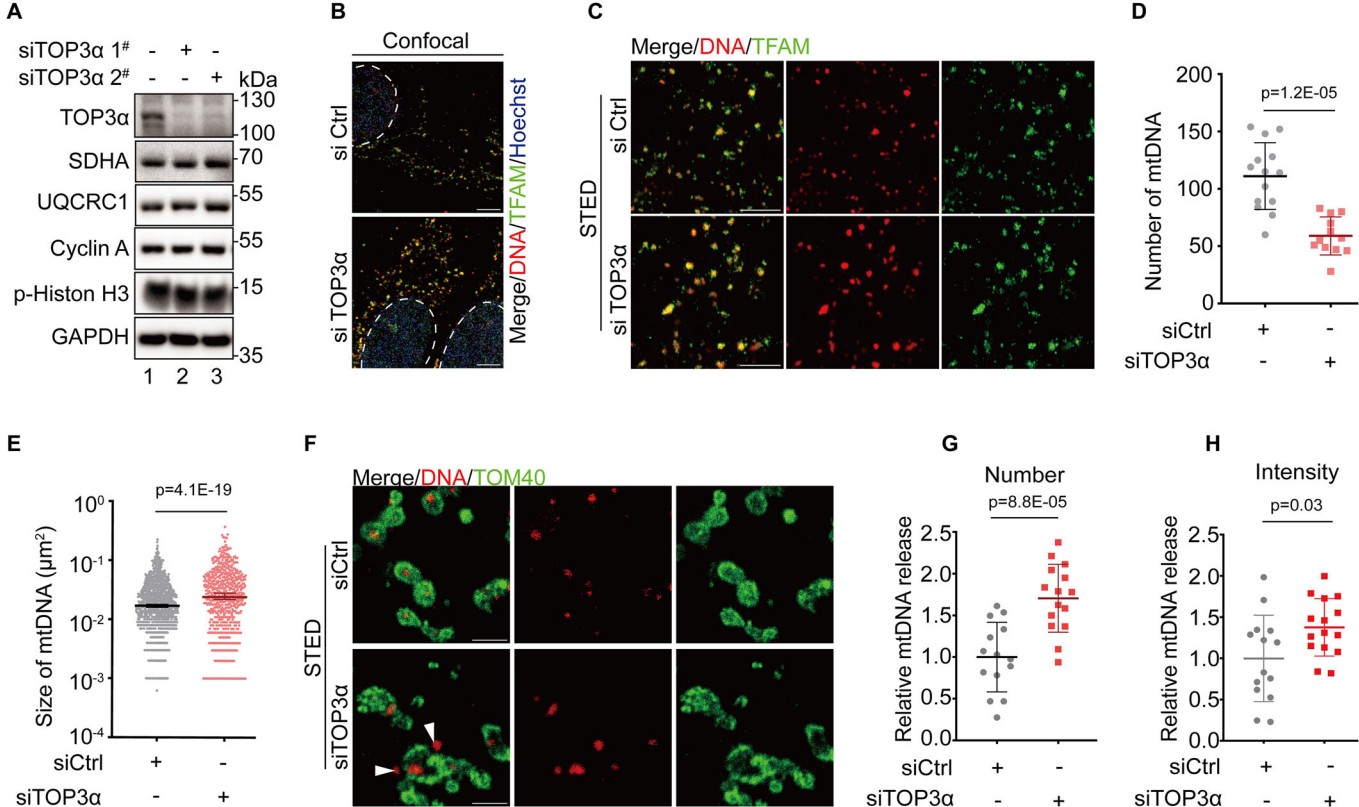

**Figure 1. TOP3α downregulation causes mtDNA clustering and release.**

(A) TOP3α level was knocked down by siRNA for 4 days in U2OS cells and subjected to SDS-PAGE and western blotting with indicated antibodies. (B) Immunofluorescence of U2OS cells following transfection of siTOP3α 2# for 4 days. Images were captured by confocal microscopy. Scale bar = 20 μm. The dotted border indicates area of nucleus. (C) U2OS cells were treated as in (B) and imaged by STED microscopy. Scale bar = 2 μm. (D) Quantification of mtDNA foci number in (C). Results were mean ± SD, n ≥ 12. Unpaired t-test was used for statistical analysis. (E) Quantification of mtDNA foci size in (C). Results were mean ± SD, n ≥ 708. Unpaired t-test was used for statistical analysis. (F) Immunofluorescence of U2OS cells following treatment as in (B). Images were captured by STED microscopy. Scale bar = 2 μm. The arrowhead indicates mtDNA outside of mitochondria. (G) The number of released mtDNA was quantified in (F). Results were mean ± SD, n ≥ 14. Unpaired t-test was used for statistical analysis. (H) The intensity of released mtDNA was quantified in (F). Results were mean ± SD, n ≥ 14. Unpaired t-test was used for statistical analysis. Source data are available online for this figure.

streptavidin pulldown, the proteins bound to mtDNA fragments were analyzed by western blotting (Fig. 3B). The cytosolic TOP3α as well as cGAS both bound to the biotin-labeled mtDNA fragments. Surprisingly, depletion of TOP3α impaired the interaction of cGAS to mtDNA without affecting the total level of cGAS in the input (Fig. 3C; Appendix Figs. S3 and S4). Moreover, direct interaction between cytosolic TOP3α and cGAS could not be detected by immunoprecipitation of cGAS (Fig. 3D). Therefore, cytosolic TOP3α promotes sensing of mtDNA by cGAS probably through modulating the conformation of released mtDNA.

The N-terminus of TOP3α serves as a mitochondrial targeting signal (MTS) (Wang et al, 2002). To understand how increased level of cytosolic TOP3α may affect the mtDNA-cGAS interaction, we deleted the MTS of TOP3α (Fig. 3E). Interestingly, deletion of the MTS did not affect the steady level of TOP3α (Fig. 3F; Appendix Fig. S2B), neither caused more mtDNA release (Appendix Fig. S5), but enhanced the expression of inflammatory genes comparing to the wild-type (WT) protein (Fig. 3G), suggesting a mitochondria-independent function of TOP3α on inflammatory regulation. Indeed, the expression of MTS-deleted version of TOP3α promoted the binding of cytosolic mtDNA to

cGAS (Fig. 3H,I). Of note, although MTS deletion increased the cytosolic level of TOP3α, the cytosolic mtDNA bound to the truncated protein was decreased, suggesting a possible competition between cGAS and TOP3α on mtDNA binding (Fig. 3J,K). To deplete cytosolic TOP3α by enhancing its mitochondrial targeting, 4xMTS derived from human COX8A was fused to the N-terminus (Appendix Fig. S6A). However, this caused the depletion of mitochondrial membrane potential (Appendix Fig. S6B,C), preventing further analysis of such construct.

## TOP3α directly facilitates cGAS binding to mtDNA

To analyze whether TOP3α could directly affect the interaction between mtDNA and cGAS, we purified a truncated version of TOP3α and the full-length cGAS (Fig. 4A–C). Subsequently, electrophoretic mobility shift assay (EMSA) was implemented to analyze DNA-protein interaction. TOP3α bound to longer mtDNA fragments more efficiently, and cGAS bound to both long and short mtDNA fragments with high efficiency (Fig. 4D,E). To analyze how TOP3α may affect the interaction between mtDNA and cGAS, biotin-labeled mtDNA fragments were incubated with purified

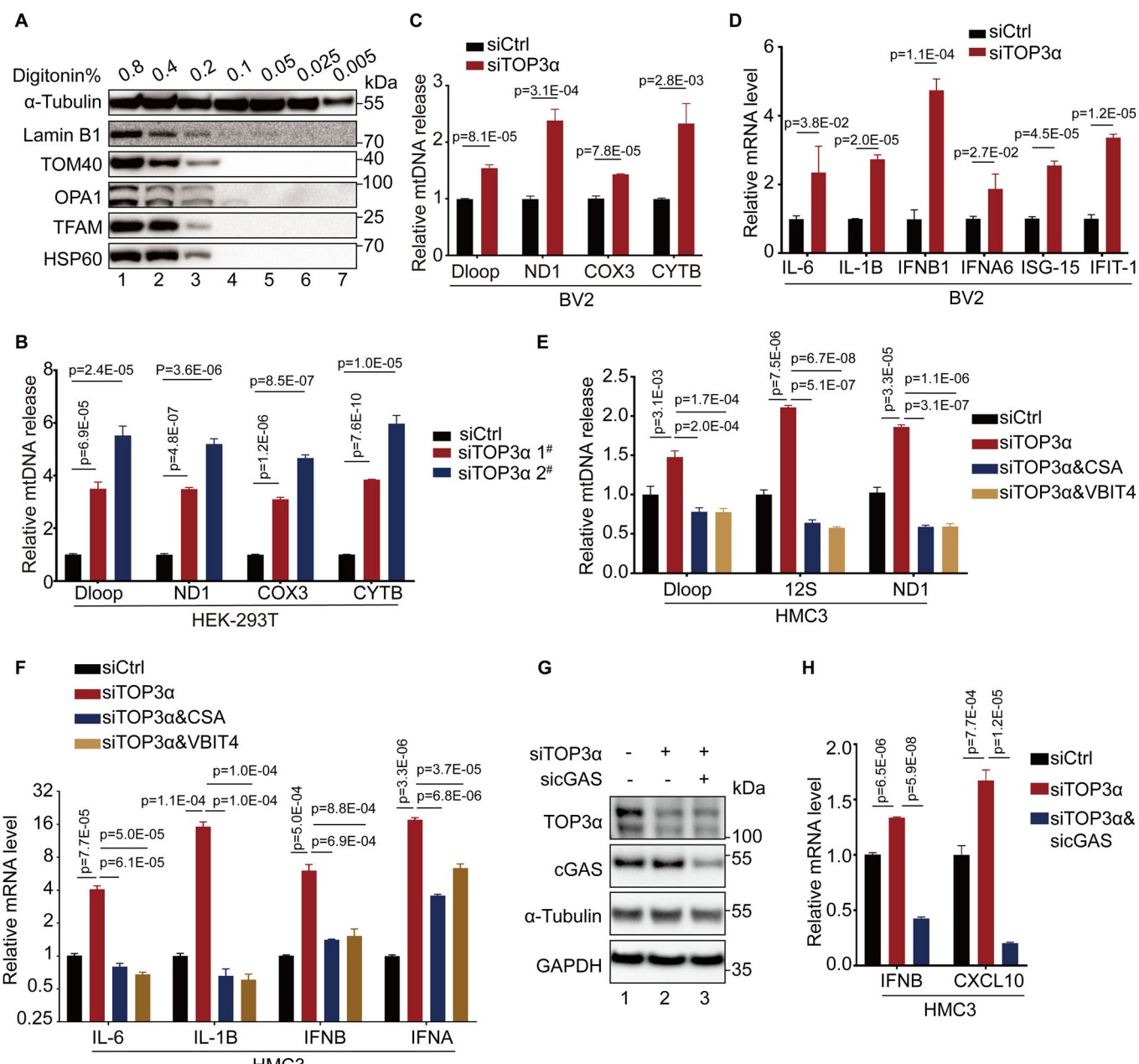

**Figure 2. Abnormal expression of TOP3α stimulates cGAS-mediated inflammation via released mtDNA.**

(A) HEK-293T cells were subjected to digitonin extraction at different concentration followed by immunoblotting with indicated antibodies. (B) HEK-293T cells were treated by TOP3α siRNA for 4 days followed by 0.025% digitonin extraction and qPCR analysis for released mtDNA. Results were mean ± SD, $n = 3$ technical replicates. Unpaired $t$-test was used for statistical analysis. (C) BV2 cells were transfected with TOP3α siRNA for 4 days and analyzed as in (B). Results were mean ± SD, $n = 3$ technical replicates. Unpaired $t$-test was used for statistical analysis. (D) BV2 cells were transfected with TOP3α siRNA for 4 days and subjected to qPCR assay for mRNA level (normalized to ACTB). Results were mean ± SD, $n = 3$ technical replicates. Unpaired $t$-test was used for statistical analysis. (E) TOP3α was knocked down by siTOP3α 2# for 4 days and treated with CsA or VBIT-4 for the last 2 days in HMC3 cells followed by qPCR analysis for released mtDNA. Results were mean ± SD, $n = 3$ technical replicates. Unpaired $t$-test was used for statistical analysis. (F) HMC3 cells were treated as in (E) and subjected to qPCR assay for mRNA level (normalized to ACTB). Results were mean ± SD, $n = 3$ technical replicates. Unpaired $t$-test was used for statistical analysis. (G) HMC3 cells were treated by TOP3α and cGAS siRNA for 4 days followed by immunoblotting with indicated antibodies. (H) HMC3 cells were treated as in (G) and subjected to qPCR analysis for mRNA level (normalized to ACTB). Results were mean ± SD, $n = 3$ technical replicates. Unpaired $t$-test was used for statistical analysis. Source data are available online for this figure.

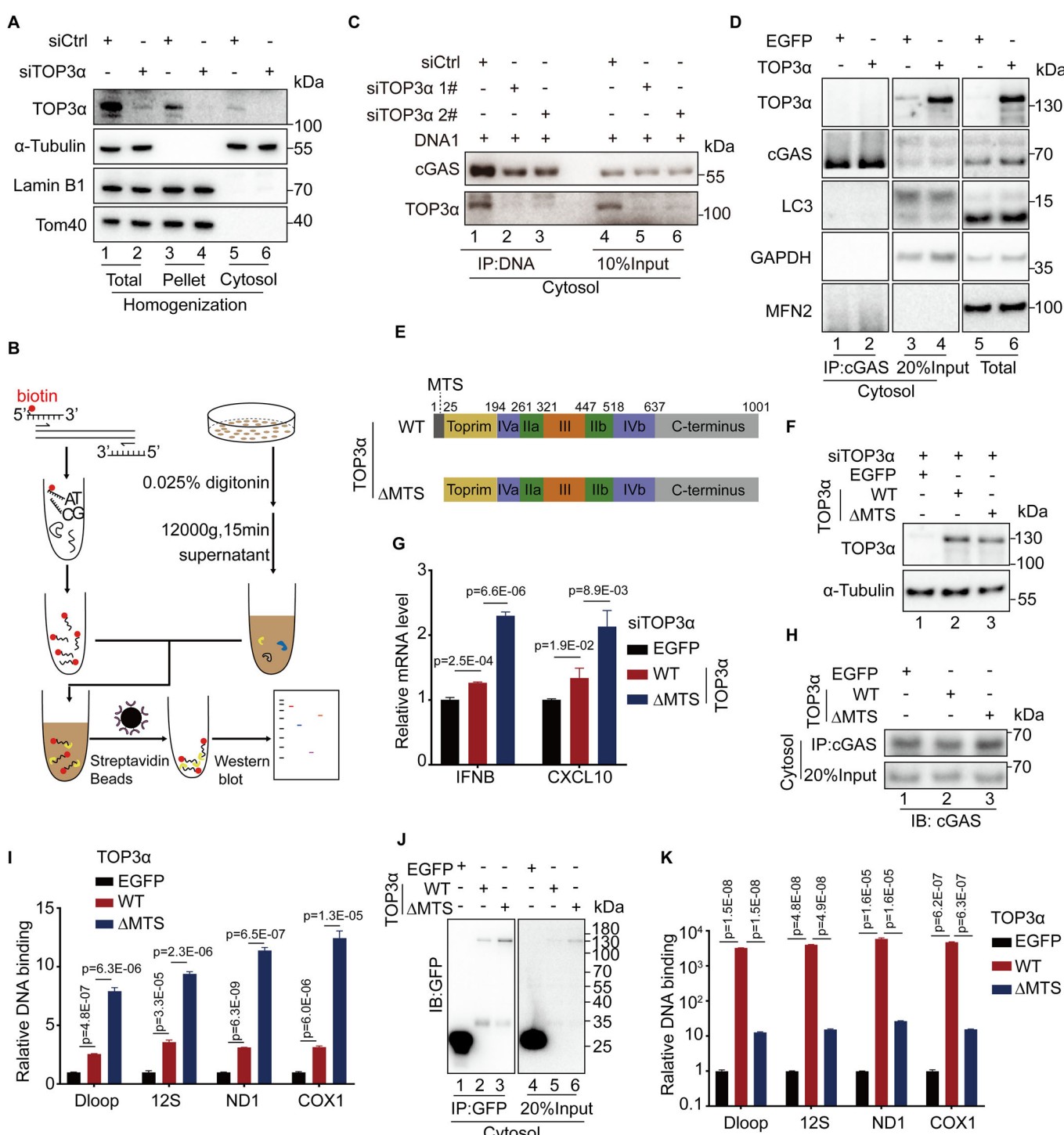

proteins before streptavidin pulldown and western blotting analysis (Fig. 4F). The presence of TOP3α promoted the binding of cGAS to mtDNA in a dose-dependent manner (Fig. 4G, compare lanes 5–6 to lane 10). Importantly, this effect was diminished when TOP3α was denatured by preheating (Fig. 4G, compare lanes 5–6 to lanes 7–8), indicating that the native conformation of TOP3α was important for this activity. Furthermore, cGAS could compete with TOP3α for binding to the mtDNA fragments (Fig. 4G, compare

lane 5 to lane 9). Taken together, cytosolic TOP3α could directly facilitate cGAS binding to released mtDNA.

## G250D mutation impairs mtDNA sensing by cGAS

Previous studies have reported that various pathogenic variants of *TOP3A* lead to the instability of mitochondrial and nuclear genomes in different clinic disorders, including Bloom syndrome-

◀
**Figure 3. Cytosolic TOP3α promotes sensing of mtDNA by cGAS.**

(A) HMC3 cells were transfected with siTOP3α 2# for 4 days before homogenization and immunoblotting analysis of different fractions. α-Tubulin is cytosolic component; Lamin B1 is nuclear component and Tom40 is mitochondrial component. (B) Scheme of interaction analysis between biotin-labeled DNA and cytosolic components. (C) U2OS cells were transfected with TOP3α siRNA for 4 days and subjected to interaction analysis as in (B). DNA1 (118 bp) was derived from 16S rRNA region of human mtDNA. (D) Co-immunoprecipitation of cytosolic components from HMC3 cells by anti-cGAS beads following transfection of plasmids encoding EGFP or EGFP tagged TOP3α for 1 day. (E) Scheme of TOP3α constructs. MTS: mitochondrial targeting signal. (F) HMC3 cells were treated by siTOP3α 2# for 4 days and transfected with plasmids encoding EGFP or siRNA-resistant EGFP tagged TOP3α constructs for the last 3 days. Whole-cell proteins were extracted and detected by immunoblotting. (G) qPCR analysis of mRNA level in HMC3 cells following treatment as in (F) (normalized to GAPDH). Results were mean ± SD, n = 3 technical replicates. Unpaired t-test was used for statistical analysis. (H) Co-immunoprecipitation of cytosolic components from HMC3 cells expressing EGFP or EGFP tagged TOP3α constructs via anti-cGAS beads. (I) qPCR analysis of mtDNA bound to endogenous cGAS following co-immunoprecipitation as in (H). Results were mean ± SD, n = 3 technical replicates. Unpaired t-test was used for statistical analysis. (J) Co-immunoprecipitation of cytosolic components from HMC3 cells expressing EGFP or EGFP tagged TOP3α constructs via anti-GFP beads. (K) qPCR analysis of mtDNA bound to EGFP or different TOP3α constructs following co-immunoprecipitation as in (J). Results were mean ± SD, n = 3 technical replicates. Unpaired t-test was used for statistical analysis. Source data are available online for this figure.

like disorder, pediatric osteosarcomas and adult-onset mitochondrial disease (de Nonneville et al, 2022; Erdinc et al, 2023; Jiang et al, 2021; Martin et al, 2018; Primiano et al, 2022). We noticed a patient with sporadic amyotrophic lateral sclerosis (ALS) enrolled at Xiangya Hospital carrying a heterozygous missense variant of *TOP3A* (c.749 G > A, p.G250D) (Liu et al, 2021). The allele frequency of G250D variant in gnomAD v4.1.0 is 2.48e-6 (4/1613174). The protein sequence alignment of multiple TOP3 homologs revealed G250 of human TOP3α as a highly conserved amino acid in a variety of evolutionarily distant species (Fig. 5A). The structure of TOP3α demonstrated that the residue G250 located at the interface of the noncatalytic CAP domain IVa and IVb (Fig. 5B) (Bocquet et al, 2014). The structure prediction indicated that G250D mutation may lead to steric clashes at the helices interface likely affecting the protein function (Fig. 5C).

Patient-derived peripheral blood mononuclear cells (PBMCs) exhibited diminished TOP3α/TFAM expression (Fig. EV5A), suggesting impaired mtDNA maintenance, corroborated by reduced mtDNA copy numbers (Fig. EV5B). Mitochondrial transcription, electron transport chain assembly, membrane potential, and ROS homeostasis remained unaffected (Fig. EV5C–F). Notably, despite increased mtDNA release (Fig. EV5G), inflammatory responses were paradoxically suppressed (Fig. EV5H), suggesting impaired mtDNA sensing for cytosolic inflammatory signaling in patient-derived cells.

To explore how TOP3α-G250D mutation impacts mtDNA sensing by cGAS, we firstly analyzed the effect of TOP3α-G250D mutation on mtDNA binding by extracting mitochondrial components and subjecting to co-immunoprecipitation (Fig. 5D,E). G250D mutation did not affect the steady level of mitochondrial TOP3α, but impaired the interaction of the mutant protein with mtDNA (Figs. 5F and EV4A–C; Appendix Fig. S7). To further analyze the effect of G250D mutation on cytosolic TOP3α, cytosolic components were extracted and subjected to co-immunoprecipitation. The cytosolic level of TOP3α was not affected by G250D mutation, but cytosolic mtDNA bound to the mutant protein was diminished (Figs. 5G,H and EV4D–F). Importantly, blue native PAGE (BN-PAGE) analysis of the cytosolic complexes indicated that G250D mutation destabilized the assembly of TOP3α polypeptide into protein complexes (Fig. 5I; Appendix Fig. S8). Taken together, TOP3α-G250D mutation impairs the protein complex assembly and mtDNA binding.

Subsequently, to analyze how TOP3α-G250D affect the sensing of released mtDNA by cGAS, biotin-labeled mtDNA fragments were transfected into cells expressing WT and mutant TOP3α followed by cytosolic extraction and streptavidin pulldown (Fig. 6A). Western blotting analysis revealed that G250D mutation not only impaired the interaction of TOP3α to mtDNA, but also diminished the mtDNA-bound cGAS proteins (Figs. 6B and EV4G; Appendix Fig. S9). Moreover, when biotin-labeled mtDNA fragments were directly incubated with cytosolic extraction, similar results were obtained (Figs. 6C,D and EV4H; Appendix Figs. S10 and S11). To obtain further evidence in this line, endogenous cGAS was co-immunoprecipitated from cytosolic extraction of cells expressing WT and mutant TOP3α. The qPCR analysis demonstrated that TOP3α-G250D expression reduced the cGAS-bound mtDNA (Fig. 6E,F). Similar observations were obtained in patient-derived PBMCs expressing TOP3α-G250D (Fig. EV5I–K). Consequently, the inflammatory responses were decreased in cells expressing TOP3α-G250D comparing to the WT (Fig. 6G). Importantly, TOP3α-G250D expression mildly affected mtDNA release comparing to the WT (Fig. 6H). Taken together, these data indicated that the G250D mutation impaired the capability of cytosolic TOP3α to promote cGAS binding of released mtDNA and the subsequent innate immune signaling.

## Discussion

Human mitochondria contain more than 1100 different proteins, 99% of which are encoded by nuclear genes (Morgenstern et al, 2021; Rath et al, 2021). Although mitochondrial genome encodes only 13 proteins of the OXPHOS system, it costs more than 20% of the mitochondrial proteome to properly maintain and express mtDNA (Gustafsson et al, 2016; Rath et al, 2021). Defects in mtDNA as well as its intricate regulatory network can disturb mitochondrial homeostasis and cause a broad range of human diseases (Suomalainen and Battersby, 2018). The change of mitochondrial membrane permeability often leads to the release of mtDNA-derived fragments into the cytosol. These fragments can serve as damage associated molecular patterns (DAMPs) to trigger innate immune responses via cytosolic nucleic acid sensors such as cGAS (West and Shadel, 2017). Importantly, mtDNA binding proteins of mitochondria are also actively involved in the regulation of innate immune signal propagation. On the one hand, depleting the abundant mitochondrial transcription factor TFAM impairs the packaging of mitochondrial nucleoids and

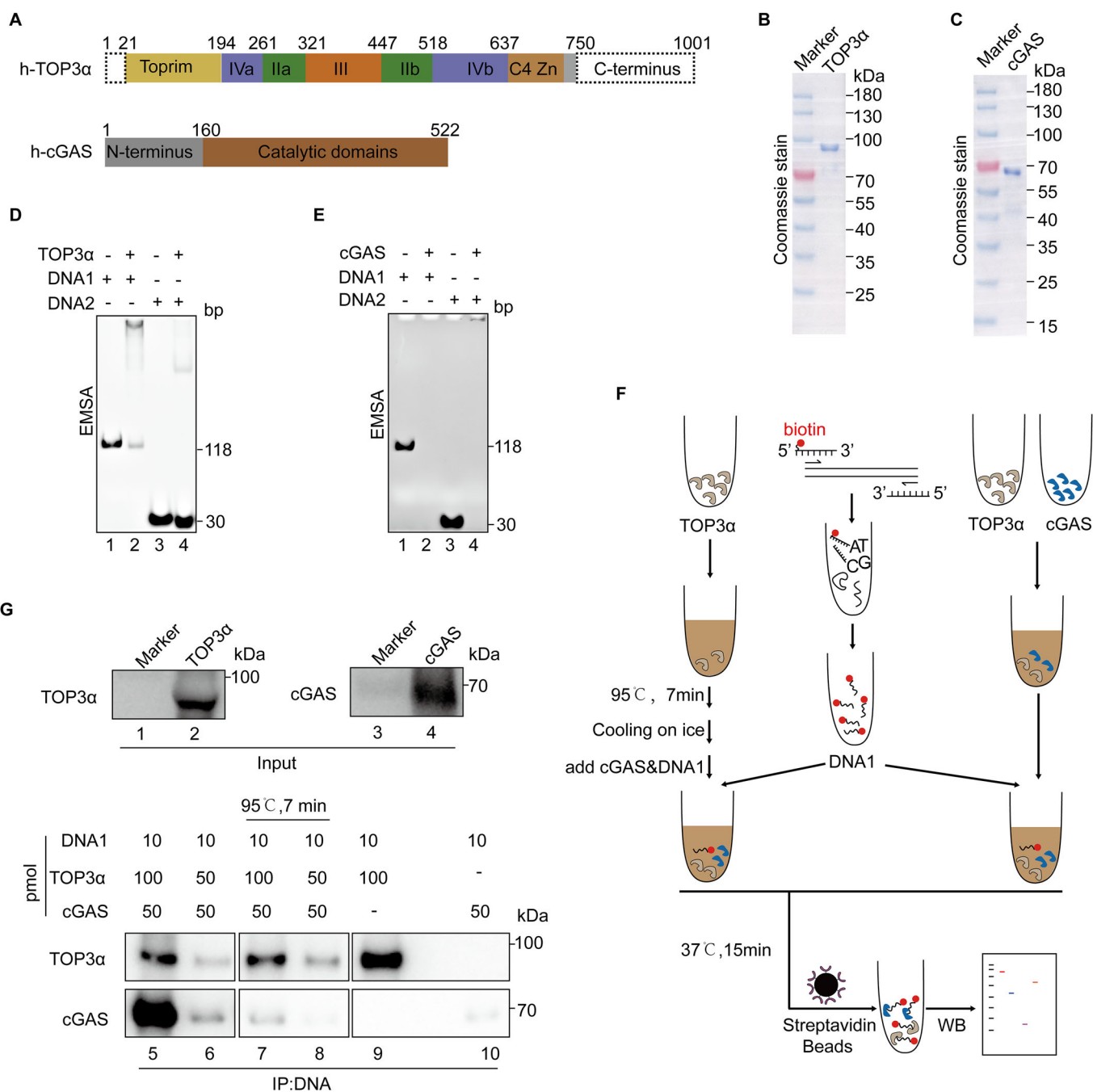

**Figure 4. TOP3α directly facilitates the interaction of mtDNA with cGAS.**

(A) Scheme of human truncated TOP3α (amino acids 21–750) and full-length cGAS constructs with 6x-His tag at the C-terminus. (B, C) Truncated TOP3α and full-length cGAS proteins were purified by Ni-NTA affinity chromatography, ion-exchange and size exclusion chromatography. 2 µg proteins were loaded for electrophoresis in SDS-PAGE and subjected to Coomassie staining. (D) Electrophoretic Mobility Shift Assay (EMSA) for the interaction of purified TOP3α proteins and mtDNA fragments. DNA1 (118 bp) and DNA2 (30 bp) were both derived from 16S rRNA region of human mtDNA. (E) Electrophoretic Mobility Shift Assay (EMSA) for the interaction of purified cGAS proteins and mtDNA fragments as in (D). (F) Scheme of interaction analysis between biotin-labeled DNA and purified proteins. (G) Purified cGAS and TOP3α proteins were subjected to interaction analysis with biotin-labeled DNA as in (F). Source data are available online for this figure.

promotes the fragmentation and release of mtDNA, thus inducing innate immune signaling (West et al, 2015). On the other hand, the released TFAM in the cytosol can impose U-turn conformation on released mtDNA to facilitate the dimerization and activation of cGAS, thus amplifying innate immune signaling (Andreeva et al, 2017; West et al, 2015). Furthermore, in the cytosol, TFAM can also function as an autophagic receptor to promote the clearance of the bound mtDNA to restrict innate immune signaling (Andreeva et al, 2017; Liu et al, 2024; West et al, 2015).

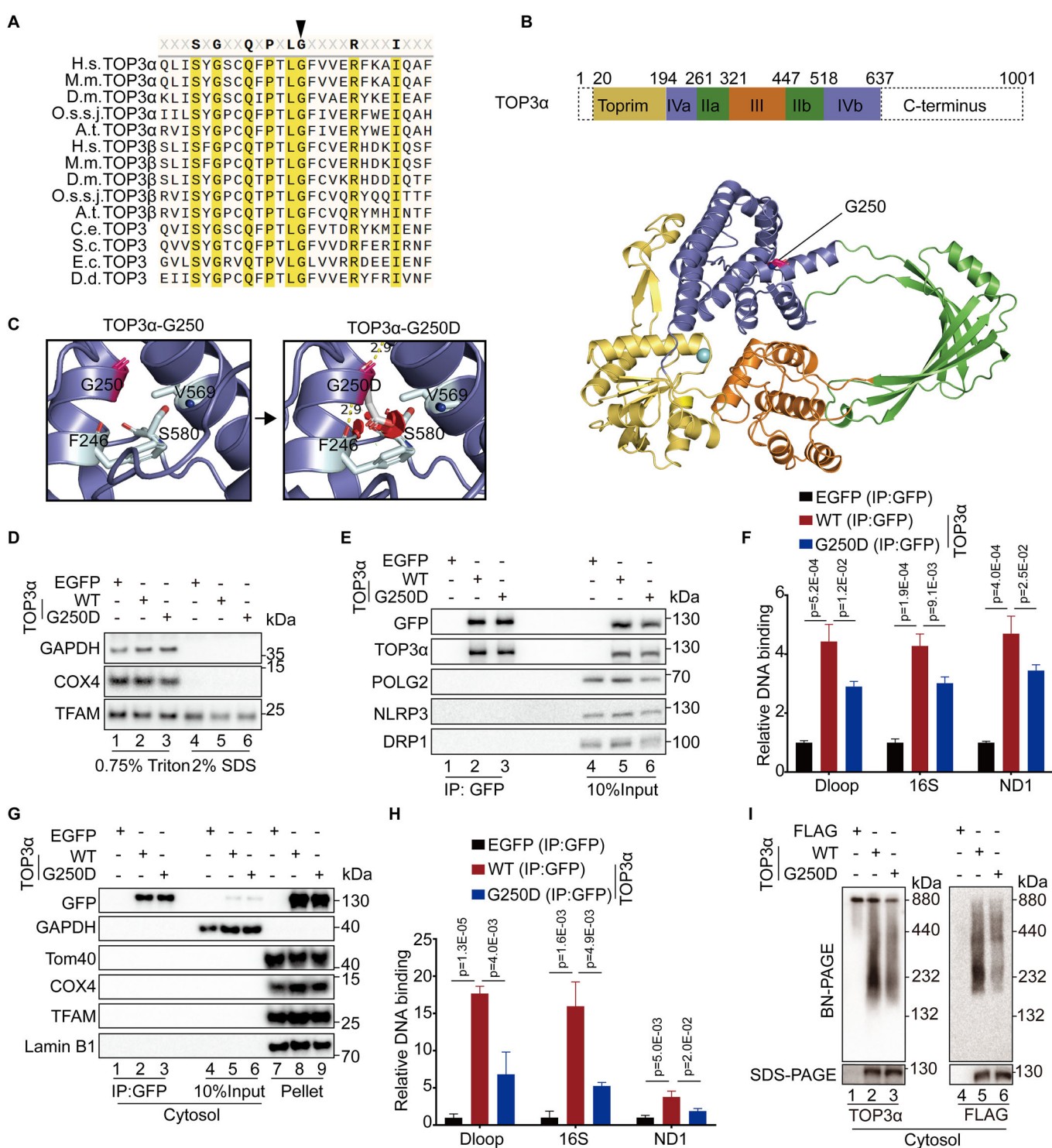

TOP3α is another important mtDNA-binding protein which plays a key role in the decatenation and segregation of mtDNA (Nicholls et al, 2018). We observed that depleting or overexpressing TOP3α could both induce aberrant mtDNA clustering, indicating that the level of TOP3α in mitochondria must be fine-tuned. Dysregulated TOP3α expression could promote mtDNA release via

mPTP-VDAC axis to stimulate cGAS-mediated innate immune responses. TOP3α has been reported to dually localize in mitochondria and nucleus (Wang et al, 2002). Intriguingly, we observed that a minor fraction of TOP3α existed in the cytosol. This was also observed by a previous study, but the function of cytosolic TOP3α was not addressed (Wang et al, 2002). Since

**Figure 5. G250D mutation impairs the binding of TOP3α to mtDNA.**

(A) Protein sequence alignment of TOP3α, TOP3β and TOP3 of different species. The arrowhead indicates G250 of human TOP3α. *Homo sapiens* (H.s.), *Mus musculus* (M.m.), *Oryza sativa subsp. japonica* (O.s.s.j.), *Arabidopsis thaliana* (A.t.), *Drosophila melanogaster* (D.m.), *Caenorhabditis elegans* (C.e.), *Saccharomyces cerevisiae* (S.c.), *Escherichia coli* (E.c.), *Dictyostelium discoideum* (D.d.). (B) The domain organization and structure of human TOP3α protein (PDB: 4CGY) showing the location of G250. (C) The predicted structure of TOP3α-G250D protein. The red sphere indicates the steric crash of G250D to the side chain of S580 and the main chain of F246. The yellow dash indicates the hydrogen bond (2.9 Å). (D) Immunoblotting of HMC3 cells expressing EGFP or EGFP tagged TOP3α for 2 days. Proteins were extracted by 0.75% Triton X-100 (supernatant) and proteins left in pellet were extracted by 2% SDS. (E) Co-immunoprecipitation of HMC3 cells following treatment as in (D). TOP3α proteins extracted by 0.75% Triton X-100 were pulled down by anti-GFP beads. (F) qPCR analysis of mtDNA bound to TOP3α following co-immunoprecipitation as in (E). Results were mean ± SD, $n = 3$ technical replicates. Unpaired *t*-test was used for statistical analysis. (G) Immunoblotting of HMC3 cells expressing EGFP or EGFP tagged TOP3α for 2 days. Cytosolic fraction was extracted by 0.025% digitonin and the pellet was extracted by 2% SDS. Cytosolic TOP3α proteins were pulled down by anti-GFP beads. (H) qPCR analysis of mtDNA bound to cytosolic TOP3α following co-immunoprecipitation as in (G). Results were mean ± SD, $n = 3$ technical replicates. Unpaired *t*-test was used for statistical analysis. (I) immunoblotting of cytosolic protein complexes extracted by 0.025% digitonin of HMC3 cells following transfection of empty vector or corresponding plasmids encoding FLAG-tagged TOP3α constructs for 2 days. Source data are available online for this figure.

TOP3α proteins are translated by the cytosolic ribosomes as other mitochondrial proteins encoded by nuclear genes, the steady level of cytosolic TOP3α is determined by the efficiency of mitochondrial and nuclear import. In addition, the release of mitochondrial TOP3α as well as the protein degradation efficiency can also influence the pool of TOP3α in the cytosol.

We investigated the function of cytosolic TOP3α and demonstrated that TOP3α in the cytosol could promote the sensing of released mtDNA by cGAS. Depleting TOP3α in the cytosol reduced the binding of cGAS to biotin-labeled mtDNA fragments. In contrast, increasing cytosolic TOP3α by expressing an MTS-deleted version of the protein enhanced cGAS-mtDNA interaction. Moreover, using an in vitro cell-free system, we unambiguously demonstrated that TOP3α could directly augment cGAS-mtDNA interaction in a dose-dependent manner. Clearly, the domain required for this function is separated from the DNA-binding region. Because the refolded protein after denaturation recovered the DNA-binding capability, but lost the function to promote cGAS-mtDNA interaction. Interestingly, our evidence did not support an interaction between cGAS and TOP3α, but rather support a competition between the two proteins on binding to mtDNA. Whether TOP3α induces particular conformation of released mtDNA to stimulate cGAS activity needs further investigation.

Pathogenic mutations of *TOP3A* have been reported in various clinical disorders (de Nonneville et al, 2022; Erdinc et al, 2023; Jiang et al, 2021; Martin et al, 2018; Primiano et al, 2022). We report here a rare mutation of a highly conserved glycine to aspartate (G250D) in the noncatalytic CAP domain of TOP3α. This mutation did not affect the steady level of TOP3α proteins, but impaired the assembly of TOP3α polypeptides into protein complexes as well as the binding to mtDNA. Moreover, G250D mutation diminished the capability of cytosolic TOP3α to augment cGAS-mtDNA interaction and the innate immune signaling. Although this mutation was first noticed in an ALS patient, more research is needed to understand the significance of TOP3α dysfunction in the disease pathogenesis. Taken together, we propose that cytosolic TOP3α facilitates innate immune responses by augmenting cGAS sensing of released mtDNA (Fig. 7). The multiple subcellular residences of TOP3α serve as an economic strategy to fulfill different cellular tasks by a single gene. Further investigation is warranted to dissect molecular mechanisms regulating the three-way tug-of-war for the final destination of a TOP3α protein.

# Methods

## Reagents and tools table

| Experimental Models | Source | Catalog Number |
|---|---|---|
| HMC3 cell line | National Collection of Authenticate Cell Culture | SCSP-5420 |
| HEK-293T cell line | National Collection of Authenticate Cell Culture | GNHu44 |
| BV2 cell line | National Collection of Authenticate Cell Culture | SCSP-5208 |
| U2OS cell line | National Collection of Authenticate Cell Culture | SCSP-5030 |
| **Recombinant DNA** | **Source** | **Catalog Number** |
| pLVX_TOP3α-EGFP | This study | N/A |
| pLVX_TOP3α-G250D-EGFP | This study | N/A |
| pLVX_TOP3α-EGFP-RNAi resist | This study | N/A |
| pLVX_TOP3α-G250D-EGFP-RNAi resist | This study | N/A |
| pLVX_ΔMTS-TOP3α-EGFP | This study | N/A |
| pLVX_4xMTS-TOP3α-EGFP | This study | N/A |
| pCW57.1_TOP3α-FLAG&HIS | This study | N/A |
| pCW57.1_TOP3α-G250D-FLAG&HIS | This study | N/A |
| pCW57.1_TOP3α-EGFP | This study | N/A |
| pCW57.1_TOP3α-G250D-EGFP | This study | N/A |
| **Antibodies** | **Source** | **Catalog Number** |
| Anti-α-Tubulin | Santa Cruz | sc-32293 |
| Anti-DNA | Progen | 61014 |
| Anti-GAPDH | Sangon | D190090 |
| Anti-Tom40 | Proteintech | 18409-1-AP |
| Anti-COX4 | CST | 4850S |
| Anti-UQCRC1 | Sangon | D123504 |
| Anti-TFAM | Proteintech | 22586-1-AP |
| Anti-POLG2 | Proteintech | 10997-2-AP |
| Anti-TOP3α | Proteintech | 14525-1-AP |
| Anti-ND6 | ABclonal | A17991 |
| Anti-SDHA | Santa Cruz | sc-390381 |
| Anti-Cyclin A | Santa Cruz | sc-271682 |
| Anti-p-Histon H3 | Santa Cruz | sc-8656-R |
| Anti-OPA1 | CST | 67589S |

| Experimental Models | Source | Catalog Number |
|---|---|---|
| Anti-GRP75 | Santa Cruz | sc-133137 |
| Anti-FLAG | Sigma | F1804 |
| Anti-GFP | Biolinkedin | L-5008 |
| Anti-Lamin B1 | ABclonal | A16685 |
| Anti-cGAS | Proteintech | 26416-1-AP |
| Anti-VDAC | Santa Cruz | sc-390996 |
| Anti-SDHC | Santa Cruz | sc-515102 |
| Anti-HSP60 | Santa Cruz | sc-13115 |
| Anti-LC3 | Proteintech | 14600-1-AP |
| Anti-MFN2 | Proteintech | 12186-1-AP |
| Anti-DRP1 | CST | 8570S |
| Anti-NLRP3 | Proteintech | 19771-1-AP |
| Peroxidase-AffiniPure Goat Anti-Mouse IgG (H + L) antibody | Jackson | 115-035-146 |
| Peroxidase-AffiniPure Goat Anti-Rabbit IgG antibody | Jackson | 111-035-144 |
| Alexa Fluor 488 goat anti-rabbit IgG | Invitrogen | A11034 |
| Alexa Fluor 647 Goat Anti-Mouse IgG | Invitrogen | A21236 |

| Oligonucleotides and other sequence-based reagents | Forward primer | Reverse primer |
|---|---|---|
| Biotin-DNA1 | CCGAAACCAGACGAG CTACCTAAGAACAGC | AACCAGCTATCAC CAGGCTCGGTAGG |
| Biotin-DNA2 | CCGAAACCAGACGAG CTACCTAAGAACAGC | GCTAGTTCTTAGGTAG CTCGTCTGGTTTCGG |
| Biotin-GFP-DNA | TCGACTTCAAGGA GGACGGCAACAT | ATCTTGAAGTTCACCTT GATGCCGTTCTTCT |
| siRNA-Homo-TOP3α 1# | CGGCUUGCCUAG UUCUCUAUU | UAGAGAACUAGGCAAGCCGTG |
| siRNA-Homo-TOP3α 2# | CAGGUUAAAGUUAAAGUUUUU | AAACUUUAACUUUA ACCUGUG |
| siRNA-Homo-TFAM 1# | GAAGAGAUAAGCAG AUUUAUU | UAAAUCUGCUUAU CUCUUCUU |
| siRNA-Homo-TFAM 2# | AAGUUCUUACCUUC GAUUUUU | AAAUCGAAGGUAAGAACUUAC |
| siRNA-Homo-cGAS | CUAUUCUCUAGCAACUUAA | UUAAGUUGCUAGAGAAUAG |
| siRNA-Murine-TOP3α | GAGUUCAACUGGA AACGAU | AUCGUUUCCAGUUGAACUC |
| TOP3α fragment (for cloning pLVX_TOP3α-EGFP) | CTGCAGTCGACGGTACCGC GGGCCCGGGATCCAATG ATCTTTCCTGTCGC CCGCTACG | TCTGTTCTGAGGACAA AAGGGACGG |
| EGFP fragment (for cloning pLVX_TOP3α-EGFP) | CCCTTTTGTCCTCAGAACA GAGGTGGCGGAGGGTCAA TGGTGAGCAAGGGCGAGG | CCCGGTAGAATTATCTA GAGTCGCGGCCGCTTACTTG TACAGCTCGTCCATGCCGAGAG |
| TOP3α-G250D fragment (for cloning pLVX_TOP3α-G250D-EGFP) | CTGCAGTCGACGGTACCGCGG GCCCGGGATCCAATGATCTTT CCTGTCGCCCGCTACG | CCACCACAAAGACC AGTGTGGG |
| TOP3α-G250D&EGFP fragment (for cloning pLVX_TOP3α-G250D-EGFP) | CACACTGGTCTTTGTGGTGGAGC | CCCGGTAGAATTATCTAGAG TCGCGGCCGCTTACTTGTACA GCTCGTCCATGCCGAGAG |
| TOP3α-RNAi resist fragment (for cloning pLVX_TOP3α-EGFP-RNAi resist) | CTGCAGTCGACGGTACCGC GGGCCCGGGATCCAATGA TCTTTCCTGTCGCCCGCTACG | TTAAATTTCAGTTTCAGTC GGTACACAGGGTGTGGCTGA CAAACTGGACACACACTGC |
| TOP3α-RNAi resist &EGFP fragment (for cloning pLVX_TOP3α-EGFP-RNAi resist) | TACCGACTGAAACTGAAAT TTAAGCGCGGTAGCCTT CCCCCGACCATGCCCTC | CCCGGTAGAATTATCTAGA GTCGCGGCCGCTTACTTGTA CAGCTCGTCCATGCCGAGAG |
| TOP3α-G250D-RNAi resist fragment (for cloning pLVX_TOP3α-G250D-EGFP-RNAi resist) | CTGCAGTCGACGGTACCGC GGGCCCGGGATCCAATGA TCTTTCCTGTCGCCCGCTACG | TTAAATTTCAGTTTCA GTCGGTACACAGGGTGT GGCTGACAAACTGG ACACACACTGC |

| Experimental Models | Source | Catalog Number |
|---|---|---|
| TOP3α-G250D-RNAi resist &EGFP fragment (for cloning pLVX_TOP3α-G250D-EGFP-RNAi resist) | TACCGACTGAAACTGAAA TTTAAGCGCGGTAGCCTT CCCCCGACCATGCCCTC | CCCGGTAGAATTATCT AGAGTCGCGGCCGCTT ACTTGTACAGCTCG TCCATGCCGAGAG |
| ΔMTS-TOP3α&EGFP fragment (for cloning pLVX_ΔMTS-TOP3α-EGFP) | CTGCAGTCGACGGTACCGC GGGCCCGGGATCCAAT GGAGATGGCCCTCCGAGGC | CCCGGTAGAATTATCTAG AGTCGCGGCCGCTTACTT GTACAGCTCGTCCATGCCGAGAG |
| 4xMTS-TOP3α&EGFP fragment (for cloning pLVX_4xMTS-TOP3α-EGFP) | CTGCAGTCGACGGTACCGC GGGCCCGGGATCCACTGGCT AGCCCACCATGTCCGTC | CCCGGTAGAATTATCTAG AGTCGCGGCCGCTTACTTGT ACAGCTCGTCCATGCCGAGAG |
| TOP3α fragment (for cloning pCW57.1_TOP3α-FLAG&HIS) | CGTCAGATCGCCTGGAGAA TTGGCTAGCATGATCTT TCCTGTCGCCCGC | TCTGTTCTGAGGAC AAAAGGGACGG |
| FLAG&HIS fragment (for cloning pCW57.1_TOP3α-FLAG&HIS) | CCGTCCCTTTTGTCCT CAGAACAGAGAATTCGG TGGCGGAGGGTCAGACTACAAAG | GAAAAGGCGCAACCCCAACC CCGGATCCTTAGTGATGATGGT GATGGTGGTGATGATG |
| TOP3α-G250D fragment (for cloning pCW57.1_TOP3α-G250D-FLAG&HIS) | CGTCAGATCGCCTGGAGAAT TGGCTAGCATGATCTTT CCTGTCGCCCGC | CCACCACAAAGACCAGTGTGGG |
| TOP3α-G250D&FLAG&HIS fragment (for cloning pCW57.1_TOP3α-G250D-FLAG&HIS) | CACACTGGTCTTTGTGGTGGAGC | GAAAAGGCGCAACCCC AACCCCGGATCCTTAGTG ATGATGGTGATGGTG GTGATGATG |
| TOP3α fragment (for cloning pCW57.1_TOP3α-EGFP) | CGTCAGATCGCCTGGAGAA TTGGCTAGCATGATCTT TCCTGTCGCCCGC | TCTGTTCTGAGGACA AAAGGGACGG |
| EGFP fragment (for cloning pCW57.1_TOP3α-EGFP) | CCCTTTTGTCCTCAGAACA GAGGTGGCGGAGGGTC AATGGTGAGCAAGGGCGAGG | GAAAAGGCGCAACCCCAA CCCCGGATCCTTACTTGTACA GCTCGTCCATGCCG |
| TOP3α-G250D fragment (for cloning pCW57.1_TOP3α-G250D-EGFP) | CGTCAGATCGCCTGGAGAAT TGGCTAGCATGATCT TTCCTGTCGCCCGC | CCACCACAAAGACCAGTGTGGG |
| TOP3α-G250D&EGFP fragment (for cloning pCW57.1_TOP3α-G250D-EGFP) | CACACTGGTCTTTGTGGTGGAGC | GAAAAGGCGCAACCCCA ACCCCGGATCCTTACTTGTA CAGCTCGTCCATGCCG |
| Homo_Dloop (for qPCR) | GAAGCAGATTTGGGTACC | TGTACTTGCTTGTAAGCATG |
| Homo_12S (for qPCR) | CCCGTTCCAGTGAGTTCACCC | CACTCTTTACGCCGG CTTCTATTGAC |
| Homo_16S (for qPCR) | AACTTTGCAAGGA GAGCCAAAGC | GGGATTTAGAGGG TTTCGTGGGC |
| Homo_ND1 (for qPCR) | ACGCCATAAAACTCTTCACCAAAG | TAGTAGAAGAGCGAT GGTGAGACTA |
| Homo_COX1 (for qPCR) | TGCCATAACCCAATACCAAACGC | CTGTTAGTAGTATAGTGA TGCCAGCAGCTAGG |
| Homo_COX2 (for qPCR) | CTACGGTCAATGCTC TGAAATCTGTG | GCTAAGTTAGCTTTA CAGTGGGCTCTAG |
| Homo_COX3 (for qPCR) | CGATACGGGATAATCC TATTTATTACCTCAG | CAGGTGATTGATAC TCCTGATGCGA |
| Homo_CYTB (for qPCR) | CGCCTGCCTGATCCTCCAA | AGGCCTCGCCCGATGTGTAG |
| Homo_GAPDH (for qPCR) | GCACCGTCAAGGCTGAGAAC | TGGTGAAGACGCCAGTGGA |
| Homo_ACTB (for qPCR) | GGCCAACCGCGAGAAGATGAC | GGATAGCACAGCCTG GATAGCAAC |
| Homo_18S (for qPCR) | GTAACCCGTTGAACCCCATT | CCATCCAATCGGTAGTAGCG |
| Homo_IFNA (for qPCR) | TGGAAGCCTGTGTGA T | ATGATTTCTGCTCTGACA |
| Homo_IFNB (for qPCR) | GTCAGAGTGGAAATCCTA AG | CTGTAAGTCTGTTAATGAAG |
| Homo_IL-6 (for qPCR) | AATAACCACCCCCTGACCCAAC | TCTGAGGTGCCCATGCTACA |
| Homo_IL-1B (for qPCR) | CAACAGGCTGC TCTGGGATT | CATGGCCACAACAACTGACG |

| Experimental Models | Source | Catalog Number |
|---|---|---|
| Homo_CXCL10 (for qPCR) | ATTTGCTGCC TTATCTTTCTG | TCTCACCCTTCTTTTTCATTGTAG |
| Murine_IFNB1 (for qPCR) | CCAGCTCCAAGA AAGGACGA | TGGATGGCAAAGGCAGTGTA |
| Murine_IL-6 (for qPCR) | CCAGAAACCGCTA TGAAGTTCC | CGGACTTGTGAAGTAGGGAAGG |
| Murine_IL-1B (for qPCR) | TTGACGGACCC CAAAAGATG | CAGCTTCTCCACAGCCACAA |
| Murine_ISG-15 (for qPCR) | CTAGAGCTAGAGCCTGCAG | AGTTAGTCACGGACACCAG |
| Murine_IFIT-1 (for qPCR) | CAAGGCAGGTTTCTGAGGAG | GACCTGGTCA CCATCAGCAT |
| Murine_ACTB (for qPCR) | CTTTTCCAGCCTTCCTTCTTGG | CGCTCAGGAGGAGCAATG |
| Murine_Dloop (for qPCR) | AGGTTTGGTCCTGGCCTTAT | GTGGCTAGGCAAGGTGTCTT |
| Murine_ND1 (for qPCR) | CTAGAAACCCCGAACCAAA | CCAGCTATCACCAAGCTCGT |
| Murine_COX3 (for qPCR) | ACCTACCAAGGCC ACCACACTCC | GCAGCCTCCTAGA TCATGTGTTGGT |
| Murine_CYTB (for qPCR) | ACAGCAAAC GGAGCCTCAA | TGCTGTGGCTATGACTGCGAACA |
| Murine_IFNA6 (for qPCR) | GCTTTCCTGATGG TTTTGGTG | AGGCTTTCTTGTTCCTGAGG |
| **Chemicals, Enzymes and other reagents** | **Source** | **Catalog Number** |
| MitoTracker Red | Invitrogen | M7512 |
| MitoSOX Red | Invitrogen | M36008 |
| Hoechst 33258 | Sigma | 14530 |
| Lipofectamine 2000 | Invitrogen | 11668019 |
| Lipofectamine RNAi MAX | Invitrogen | 13778150 |
| Phanta Max Super-Fidelity DNA Polymerase | Vazyme | P505 |
| pEASY-Basic Seamless Cloning and Assembly Kit | TransGen Biotech | CU201 |
| Protease Inhibitor Cocktail | Sigma | 11873580001 |
| RevertAid Master Mix | Thermo Fisher Scientific | M1631 |
| SYBR qPCR SuperMix | Novoprotein | E096 |
| Total RNA Extraction Reagent | Vazyme | R401-01 |
| Proteinase K | Coolaber | CP9191 |
| Streptavidin beads | Thermo | 20353 |
| EMSA/Gel-Shift binding buffer | Beyotime | GS005 |
| EMSA/Gel-Shift loading buffer | Beyotime | GS007 |
| EMSA Precast PAGE gel | Beyotime | GS306S |
| TBE buffer | Beyotime | R0223 |
| GelRed dye | Vazyme | GR501 |
| **Software** | **Source** | |
| GraphPad Prism v7.0 | Dotmatics | https://www.graphpad.com/ |
| Snapgene v4.2.4 | Dotmatics | https://www.snapgene.com/ |
| Fiji | ImageJ | https://imagej.net/software/fiji/ |
| **Other** | **Source** | |
| Axio Imager M2 microscope | ZEISS | |
| STEDYCON microscope | Abberior Instruments | |
| HIS-SIM microscope | CSR Biotech | |
| ChampChemi 910 imager | Sagecreation | |

## Ethics approval and consent to participate

The study protocol was approved by the Ethics Committee and the Expert Committee of Xiangya Hospital, Central South University (CSU) (Ethics Approval No. 2023111019). Written informed consent was obtained from all enrolled participants.

## Cell culture and transfection

HEK-293T, U2OS, HMC3 and BV2 cells were obtained from the National Collection of Authenticate Cell Culture. The cultivation of HEK-293T, U2OS and HMC3 cells was performed in DMEM medium, supplemented with 10% fetal bovine serum and 1% penicillin-streptomycin. The BV2 cells were cultured in RPMI-1640 medium. For transfection, 3 µg of plasmids and 80 pmol of siRNAs (Reagents and Tools Table) per well of a 6-well plate were transfected using Lipofectamine 2000 (Invitrogen, 11668019) and Lipofectamine RNAi MAX (Invitrogen, 13778150), respectively, in accordance with standard procedures. For mitochondrial staining, 100 nM MitoTracker Red (Invitrogen, M7512) or 5 µM MitoSOX Red (Invitrogen, M36008) was dissolved in medium and stained for 30 min.

## Plasmid construction

The human *TOP3A* cDNA template was obtained from WZ Biosciences (CH897424). The cDNA was amplified by Phanta Max Super-Fidelity DNA Polymerase (Vazyme, P505) and cloned into the pLVX or pCW57.1 vector fused with EGFP or FLAG tag at the C-terminus of the gene using the pEASY-Basic Seamless Cloning and Assembly Kit (TransGen Biotech, CU201) in accordance with the standard protocol. The truncated TOP3α (ΔMTS), G250D or RNAi-resistant version of TOP3α were amplified with corresponding primers for seamless cloning (Reagents and Tools Table). All plasmids were verified by sequencing in Sangon Biotech.

## Western blotting

For SDS-PAGE, the SDS sample buffer (40 mM Tris-HCl pH 7.4, 100 mM NaCl, 20% glycerol, 0.2 mM EDTA, 2% SDS) supplemented with 1 × complete EDTA-free Protease Inhibitor Cocktail (Sigma, 11873580001) and 1 mM PMSF was used for the extraction of whole-cell proteins before denaturing at 95 °C for 10 min. For BN-PAGE, proteins were extracted using the digitonin lysis buffer (1% digitonin, 20 mM Tris-HCl pH 7.4, 0.1 mM EDTA, 150 mM NaCl, 10% glycerol) supplemented with 1 × Proteinase Inhibitor and 1 mM PMSF. The bicinchoninic acid (BCA) assay was then performed to determine the protein concentration. After electrophoresis, the proteins were transferred onto PVDF membranes before immunodecoration with antibodies listed in Reagents and Tools Table.

## Immunofluorescence

The cells were seeded on coverslips (Assistant, 41001115) followed by fixation with 4% paraformaldehyde at room temperature for 15 min. The fixed cells were permeabilized with 0.1% Triton X-100 and blocked with 2% BSA, followed by incubation with primary

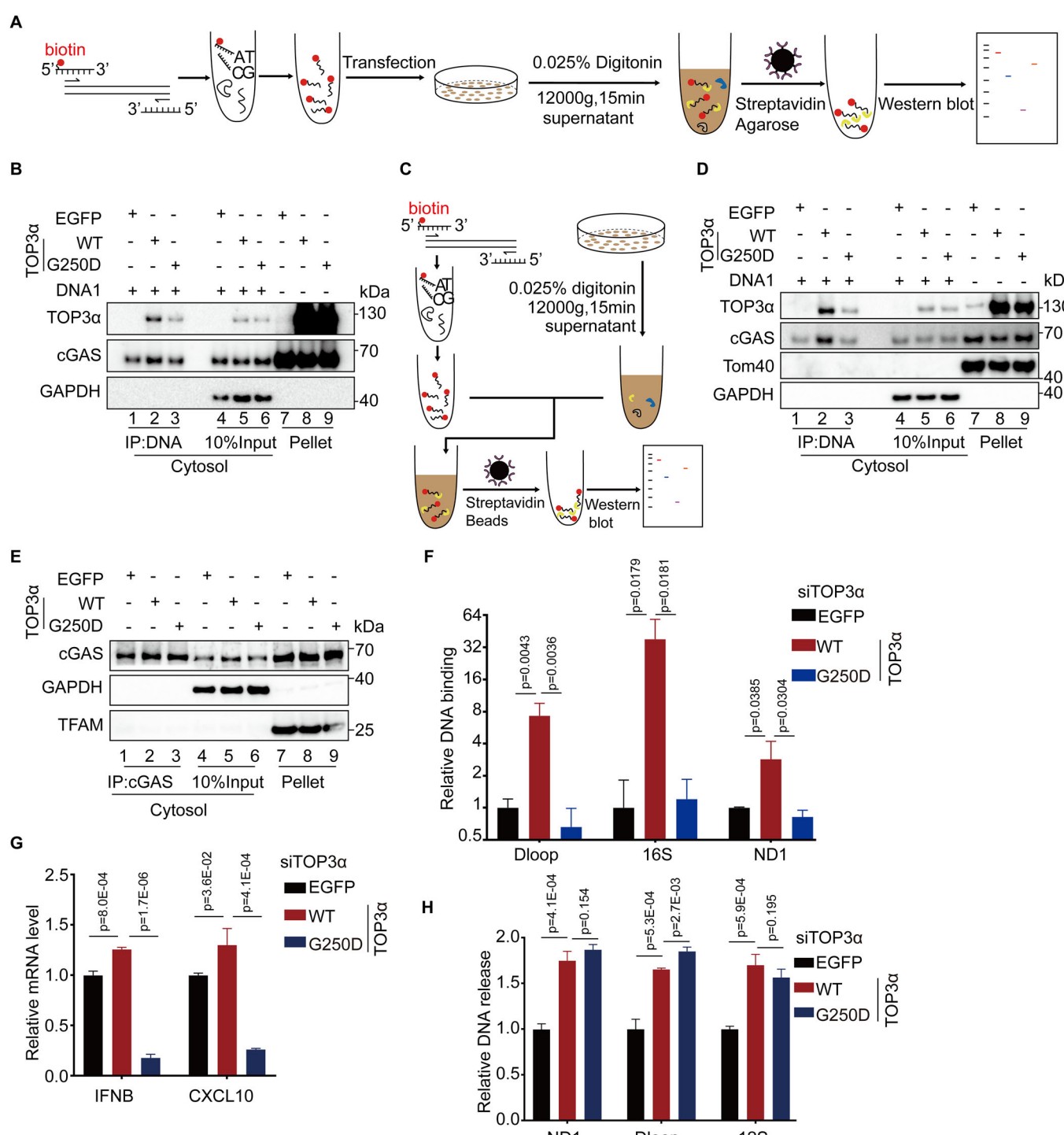

**Figure 6. TOP3α-G250D suppresses the interaction of cGAS with mtDNA.**

(A) Overview of interaction analysis between cytosolic components and transfected biotin-labeled DNA. (B) The biotin-labeled DNA was transfected into HMC3 cells for pull down of cytosolic components by streptavidin beads as in (A). HMC3 cells were transfected with empty vector or corresponding plasmids encoding EGFP tagged TOP3α constructs for 2 days. (C) Overview of interaction analysis between biotin-labeled DNA and cytosolic components. (D) HMC3 cells expressing EGFP or EGFP tagged TOP3α for 2 days and subjected to interaction analysis as in (C). (E) HMC3 cells were treated by TOP3α siRNA 2# for 4 days and transfected with plasmids encoding EGFP or siRNA-resistant EGFP tagged TOP3α constructs for the last 3 days. Co-immunoprecipitation of cytosolic components via anti-cGAS beads was analyzed by SDS-PAGE and immunoblotting. (F) qPCR analysis of mtDNA bound to the endogenous cGAS following co-immunoprecipitation in (E). Results were mean ± SD, $n = 3$ technical replicates. Unpaired $t$-test was used for statistical analysis. (G) HMC3 cells were treated as in (E), followed by qPCR analysis for mRNA level (normalized to GAPDH). Results were mean ± SD, $n = 3$ technical replicates. Unpaired $t$-test was used for statistical analysis. (H) HMC3 cells were treated as in (E) followed by qPCR analysis for mtDNA release. Results were mean ± SD, $n = 3$ technical replicates. Unpaired $t$-test was used for statistical analysis. Source data are available online for this figure.

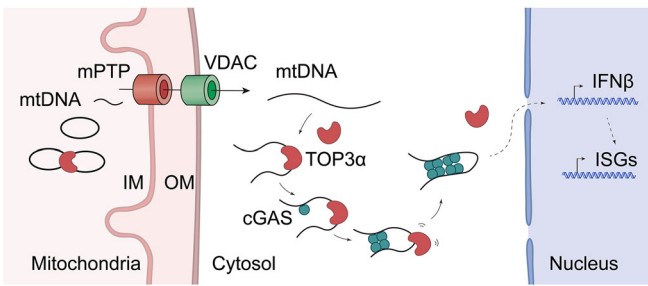

**Figure 7. Proposed model of TOP3α function.**

Fragments of mtDNA are released into cytosol via mPTP and VDAC. Cytosolic TOP3α binds to released mtDNA and promotes the engagement of cGAS to induce the inflammatory responses.

antibodies at 4 °C overnight and with Alexa Fluor-conjugated secondary antibodies at room temperature for 1 h (Reagents and Tools Table). Prior to mounting, the cells were stained with 10 µg/mL Hoechst 33258 (Sigma, 14530) for 5 min. Imaging was performed using the Axio Imager M2 (ZEISS), the STEDYCON (Abberior Instruments) or the HIS-SIM (CSR Biotech) microscopes. Number of mtDNA was quantified by ImageJ following the instruction of particle analysis.

## Total RNA extraction and real-time quantitative polymerase chain reaction (RT-qPCR)

Total RNA was extracted using the Total RNA Extraction Reagent (Vazyme, R401-01), followed by reverse transcription with RevertAid Master Mix (Thermo Fisher Scientific, M1631) in accordance with the manufacturer's instructions. The primers utilized for RT-qPCR are enumerated in Reagents and Tools Table. The reactions were prepared using the SYBR qPCR SuperMix (Novoprotein, E096). All reactions were conducted in triplicate in accordance with standard procedures.

## Quantification of released mtDNA

The released mtDNA was measured as previously described (Kim et al, 2019; Yu et al, 2020). 300 µL digitonin buffer (0.025% digitonin in DPBS) was used to extract cytosolic components of cells ($2 \times 10^6$ cells) at 4 °C for 15 min, followed by centrifugation at $13,000 \times g$ for 15 min at 4 °C. The supernatant was used for qPCR directly. The pellet was lysed in 300 µL lysis buffer (2% SDS, 1 mM EDTA) supplemented with 100 µg/mL proteinase K (Coolaber, CP9191) and incubated at 55 °C overnight. The proteinase K was denatured at 95 °C for 20 min. The pellet was diluted with water (1:100) and added into qPCR assay with mtDNA-specific primers (Reagents and Tools Table). The released mtDNA was normalized to total mtDNA in pellet for each sample.

## DNA oligo preparation

The 30-nt forward-strand DNA and their corresponding complementary strand of 16S rRNA coding region in human mtDNA were synthesized, purified by HPLC, and annealed to form duplex DNA (DNA2) by GenScript Biotech (Nanjing, China). The 118-bp

dsDNA of 16S rRNA coding region in mtDNA (DNA1) was amplified by PCR using the human mtDNA as a template. The forward and reverse primers are shown in Reagents and Tools Table. The PCR product was pooled and further purified by ion-exchanged chromatography using 5-ml HiTrap™ Q HP column (Cytiva, USA) with the use of a linear gradient of 0–1600 mM NaCl in 25 mM Tris-HCl pH 7.5. The DNA purity was verified by agarose gel electrophoresis. The DNA-containing fractions were pooled, concentrated using the Amicon Ultra centrifugal filter unit 15 mL - 3 kDa molecular-weight cutoff (Merck Millipore, USA), and stored at −80 °C until use.

## Recombinant protein expression and purification

The human cDNAs encoding full-length cGAS (amino acids 1–522) and truncated TOP3α (amino acids 21–750) were synthesized by GenScript Biotech (Nanjing, China) with sequence optimized for *E. coli* overexpression. Both full-length cGAS and truncated TOP3α cDNAs were cloned into pET-21a (+) vector generating a two-amino-acid (LE) cloning artifact and 6x-His tag at the C-terminus, and were expressed in *E. coli* Transetta (DE3) cells and BL21 (DE3) pLysS cells, respectively.

For TOP3α purification, cells were homogenized in lysis buffer containing 50 mM Tris-HCl pH 8.5, 250 mM NaCl, 1 mM PMSF, 5% (v/v) glycerol and 10 µg/ml DNase I, centrifuged at $14,000 \times g$ for 1 h at 4 °C and then applied onto a Protino Ni-NTA agarose beads (Macherey-Nagel, Germany) for affinity chromatography following the manufacturer's instructions. The column was washed thoroughly with equilibration buffer (20 mM Tris–HCl pH 8.5, 250 mM NaCl and 10% glycerol) followed by 10 mM imidazole in equilibration buffer. Proteins were eluted with 250 mM imidazole in equilibration buffer, and were further desalted by HiPrep 26/10 Desalting column (Cytiva, USA) connected to UEV25 purification system (Union-Biotech, China), using the running buffer of 50 mM Tris-HCl pH 8.5, 30 mM NaCl and 10% glycerol, followed by ion-exchanged chromatography using 5-ml HiTrap Q HP column (Cytiva, USA) with the use of a linear gradient of 30–800 mM NaCl. After the purity of proteins had been analyzed by SDS-PAGE, the TOP3α-containing fractions were pooled, concentrated using the Amicon Ultra centrifugal filter unit 15 mL - 10 kDa molecular-weight cutoff (Merck Millipore, USA) and then further purified by size-exclusion chromatography on a HiLoad 16/600 Superdex 200 pg (Cytiva, USA) in a running buffer containing 50 mM Tris-HCl pH 7.5, 150 mM NaCl, 0.5 mM TCEP and 10 mM MgCl$_2$.

For cGAS purification, cells were homogenized in lysis buffer containing 25 mM HEPES pH 7.5, 250 mM NaCl, 1 mM PMSF, 5% (v/v) glycerol and 10 µg/ml DNase I, centrifuged and applied onto a Protino Ni-NTA affinity column operated at 4 °C, followed by the addition of 20 column volumes of equilibration buffer (25 mM HEPES pH 7.5, 120 mM NaCl), and then 7 column volumes of equilibration buffer containing 10 mM imidazole. The proteins were then eluted with 25 mM HEPES pH 7.5, 120 mM NaCl, 0.5 mM TCEP, 10% glycerol containing 250 mM imidazole. Fractions with cGAS were then further desalted by HiPrep 26/10 Desalting column connected to UEV25 purification system, followed by ion exchanged chromatography using 5-ml HiTrap SP HP column (Cytiva, USA) with the use of a linear gradient of 30–1200 mM NaCl. cGAS protein was further purified by size-

exclusion chromatography on a HiLoad 16/600 Superdex 200 pg in a running buffer containing 25 mM HEPES pH 7.5, 120 mM NaCl and 0.5 mM TCEP. Purified proteins were concentrated, flash-frozen in liquid nitrogen, and stored at −80 °C for biochemical experiments.

## Immunoprecipitation and pull-down assay

Cytosolic components were extracted with the digitonin buffer (50 mM Tris-HCl pH 7.5, 150 mM NaCl and 0.025% digitonin) supplemented with 1 × complete EDTA-free Protease Inhibitor Cocktail (Sigma, 11873580001) on ice for 15 min. The extracted cell lysates were then subjected to two subsequent centrifugation steps at $12,000 \times g$ for 15 min at 4 °C. The supernatants were taken for immunoprecipitation with anti-FLAG (Beyotime, P2115), anti-GFP (AlpalifeBio, KTSM1334) or Streptavidin (Thermo, 20353) beads. The bound proteins were analyzed by SDS-PAGE and immuno-blotting with indicated antibodies. To detect the level of protein-bound mtDNA, the immunoprecipitants were digested with 100 μg/ml proteinase K overnight, followed by qPCR assay with mtDNA-specific primers (Reagents and Tools Table). The level of protein-bound mtDNA was normalized to the level of mtDNA from the input and the level of immunoprecipitated proteins.

For immunoprecipitation in an in vitro cell-free system, purified cGAS, TOP3α and biotin-labeled DNA were incubated in the buffer containing 50 mM Tris-HCl pH 7.5 and 150 mM NaCl at 37 °C for 15 min and at room temperature for 2 h, then followed by immuno-precipitation with streptavidin beads, SDS-PAGE and western blotting.

## Electrophoretic mobility-shift assays (EMSA)

EMSA was performed as previously described (Hellman and Fried, 2007). Briefly, 5 pmol DNA1, 50 pmol DNA2 and 300 pmol TOP3α or cGAS were used in 10 μL reaction. The mixture was incubated at 37 °C for 15 min in EMSA/Gel-Shift binding buffer (Beyotime, GS005), followed by adding EMSA/Gel-Shift loading buffer (Beyotime, GS007). All samples were loaded into 6% EMSA Precast PAGE gel (Beyotime, GS306S) and subjected to electrophoresis in TBE buffer (Beyotime, R0223) at 4 °C. The gel was stained with 3x GelRed dye (Vazyme, GR501) prepared in 0.5x TBE buffer and images were acquired using ChampChemi 910 (Sagecreation).

## Statistical analysis

GraphPad Prism 8 was employed for statistical analysis. Significance between two groups was determined by the unpaired t-test. Fisher's exact test was used to determine the association between two categorical variables. $P < 0.05$ was considered to indicate statistical significance.

# Data availability

This study includes no data deposited in external repositories.

The source data of this paper are collected in the following database record: biostudies:S-SCDT-10_1038-S44319-025-00614-2.

# Peer review information

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

## Acknowledgements

We thank Guangzhou CSR Biotech Co. Ltd for imaging by using their commercial super-resolution microscope (HIS-SIM), data acquisition, SR image reconstruction. The STED images were acquired using Abberior STEDYCON (Abberior Instruments GmbH, Göttingen, Germany) fluorescence microscope. This work was supported by grants from the National Nature Science Foundation of China (32270722, 82371445, 82171431), the National Health Commission of China (2024ZD0530500), the Ministry of Science and Technology of China (2021YFA0805202, 2021ZD0201803), the Natural Science Foundation of Hunan Province (2022JJ30914), the Department of Science & Technology of Hunan Province (2019RS1010), Outstanding Postdoctoral Innovative Talent Project of Hunan Province (2021RC2037) and Central South University (2023QYJC035), China Postdoctoral Science Foundation (2020M672515).

## Author contributions

**Dongjing Cai**: Conceptualization; Resources; Data curation; Software; Formal analysis; Validation; Investigation; Visualization; Methodology; Writing—original draft; Writing—review and editing. **Cheng Chen**: Conceptualization; Resources; Data curation; Software; Formal analysis; Validation; Investigation; Visualization; Methodology; Writing—review and editing. **Piyanat Meekrathok**: Resources; Data curation; Formal analysis; Funding acquisition; Methodology; Writing—review and editing. **Weiqian Zeng**: Software; Funding acquisition; Methodology; Writing—review and editing. **Zheng Wang**: Data curation; Software; Formal analysis; Methodology; Writing—review and editing. **Zhigang Peng**: Software; Formal analysis; Methodology; Writing—review and editing. **Yunan Mo**: Formal analysis; Methodology; Writing—review and editing. **Xia Xu**: Funding acquisition; Validation; Methodology; Writing—review and editing. **Junling Wang**: Resources; Software; Funding acquisition; Methodology; Writing—review and editing. **Jian Qiu**: Conceptualization; Resources; Supervision; Funding acquisition; Writing—original draft; Project administration; Writing—review and editing.

Source data underlying figure panels in this paper may have individual authorship assigned. Where available, figure panel/source data authorship is listed in the following database record: biostudies:S-SCDT-10_1038-S44319-025-00614-2.

## Disclosure and competing interests statement

The authors declare no competing interests.

# Expanded View Figures

**Figure EV1.  Knocking-down TOP3α causes mtDNA clustering and release.**

(A) TOP3α level was knocked down by siRNA for 4 days in U2OS cells followed by immunofluorescence. Scale bar = 20 μm. (B) Immunofluorescence of U2OS cells following treatment as in (A). White squared area was zoomed in. Images were captured by Apotome microscopy. Scale bar (overview) = 20 μm, scale bar (zoom) = 2 μm. (C) Immunofluorescence of U2OS cells following treatment as in (A). White squared area was zoomed in. The released mtDNA was marked by white arrowhead. Scale bar (overview) = 20 μm, scale bar (zoom) = 2 μm. (D) The number of released mtDNA was quantified in (C). Results were mean ± SD, *n* = 7. Unpaired *t*-test was used for statistical analysis. (E) The intensity of released mtDNA was quantified in (C). Results were mean ± SD, *n* = 7. Unpaired *t*-test was used for statistical analysis. (F) Immunofluorescence of U2OS cells following treatment as in (A). White squared area was zoomed in. The released mtDNA was marked by white arrowhead. Images were captured by HIS-SIM microscopy. Scale bar (overview) = 20 μm, scale bar (zoom) = 2 μm. (G) Plot of immunofluorescent intensity of DNA and Tom40 along the white line drawn in (F).

▶

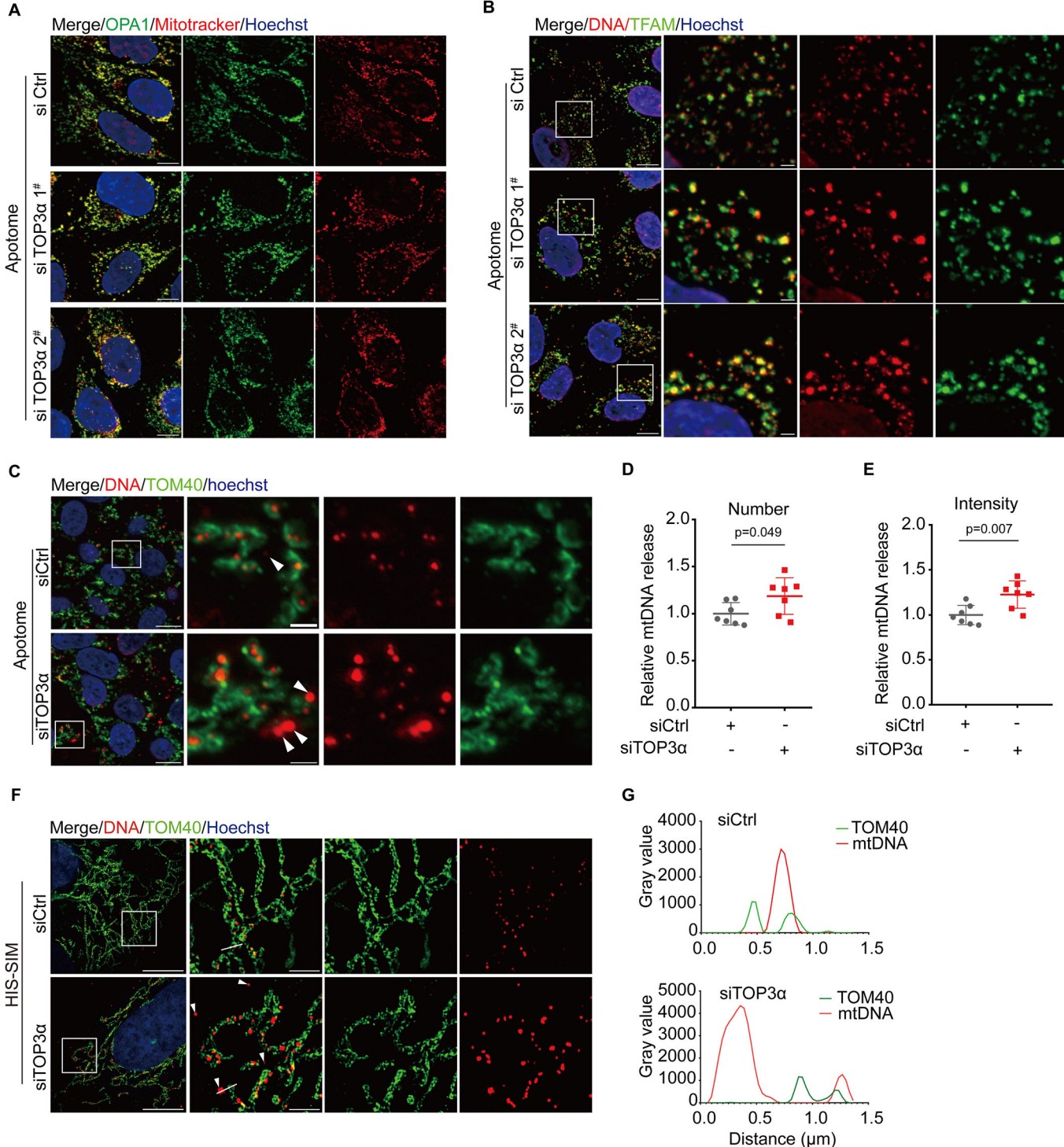

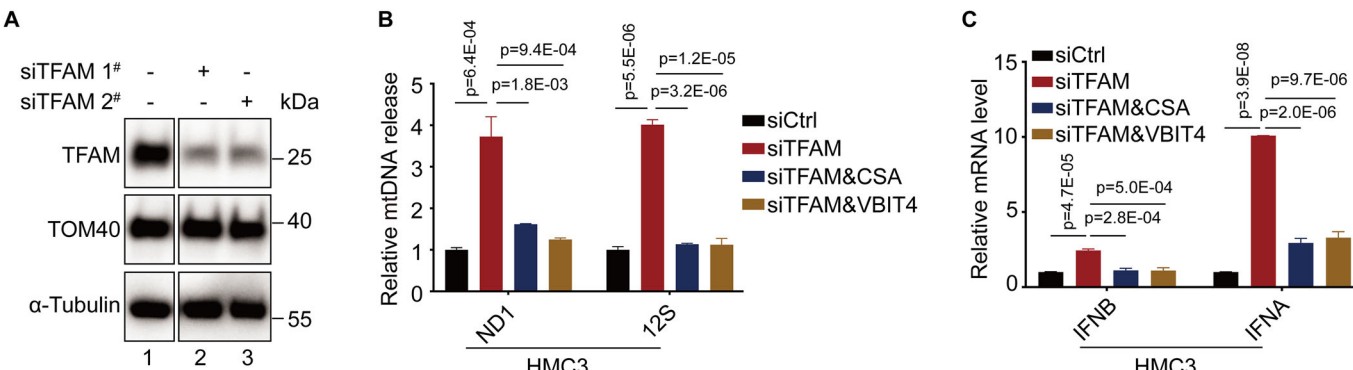

**Figure EV2. Knocking-down TFAM induces mtDNA release and inflammation.**

(A) Depletion of TFAM by siRNA followed by immunoblotting with indicated antibodies. (B) TFAM was knocked down by siRNA in HMC3 cells for 4 days and treated with CsA or VBIT-4 for the last 2 days followed by qPCR analysis for released mtDNA. Results were mean ± SD, $n = 3$ technical replicates. Unpaired $t$-test was used for statistical analysis. (C) HMC3 cells were treated as in (B) and subjected to qPCR assay for mRNA level (normalized to ACTB). Results were mean ± SD, $n = 3$ technical replicates. Unpaired $t$-test was used for statistical analysis.

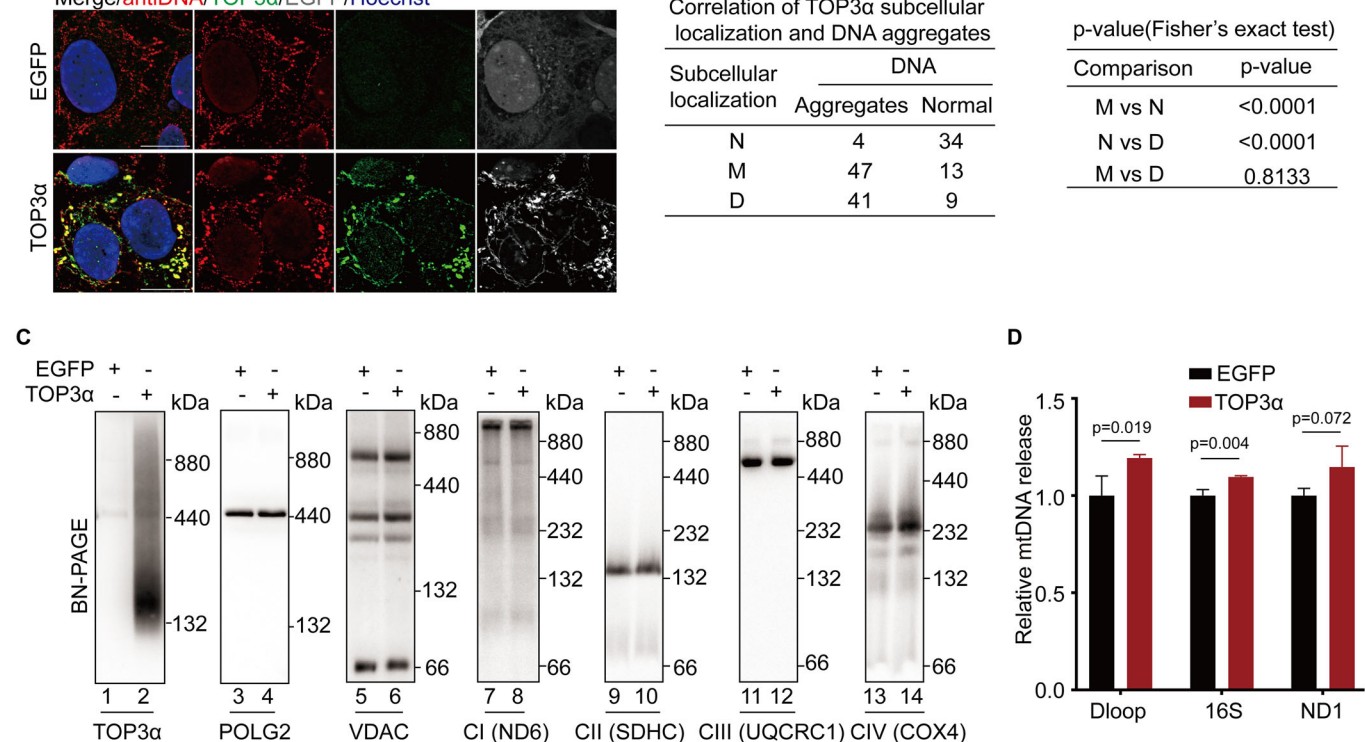

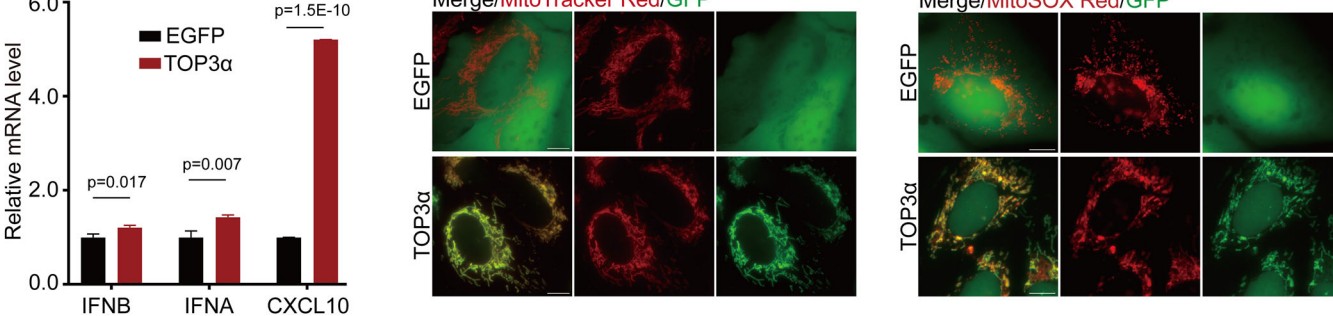

**Figure EV3. TOP3α overexpression leads to mtDNA aggregation and inflammatory response.**

(A) Immunofluorescence of U2OS cells expressing EGFP or EGFP tagged TOP3α for 3 days, scale bar = 20 μm. (B) Correlation of TOP3α subcellular localization and mtDNA aggregation. N: nucleus; M: mitochondria; D: dual localization. Fisher's exact test was used for statistical analysis. (C) Proteins of U2OS cells expressing EGFP or EGFP tagged TOP3α for 3 days were extracted by 1% digitonin and separated by BN-PAGE followed by immunoblotting with indicated antibodies. (D) U2OS cells expressing EGFP or EGFP tagged TOP3α for 3 days were analyzed by qPCR for cytosolic mtDNA with indicated primers. Results were mean ± SD, $n = 3$ technical replicates. Unpaired $t$-test was used for statistical analysis. (E) U2OS cells expressing EGFP or EGFP tagged TOP3α for 3 days were analyzed by qPCR for indicated mRNA (normalized to ACTB). Results were mean ± SD, $n = 3$ technical replicates. Unpaired $t$-test was used for statistical analysis. (F) U2OS cells expressing EGFP or EGFP tagged TOP3α for 3 days were stained with MitoTracker Red for live-cell imaging. Scale bar = 20 μm. (G) U2OS cells expressing EGFP or EGFP tagged TOP3α for 3 days were stained with MitoSOX Red for live-cell imaging. Scale bar = 20 μm.

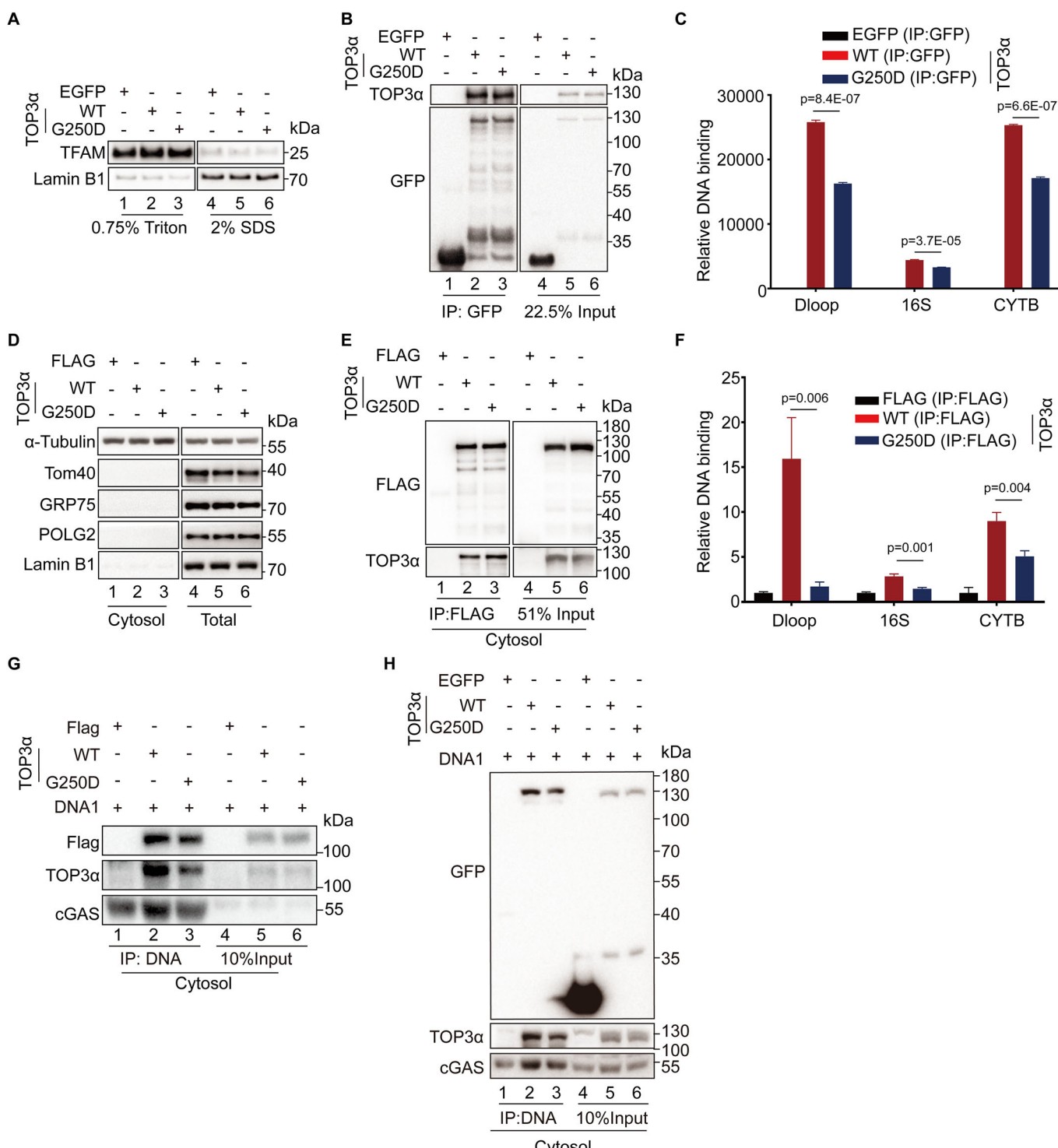

**Figure EV4. G250D mutation impairs the binding of TOP3α and cGAS to mtDNA.**

(A) Immunoblotting of U2OS cells expressing EGFP or EGFP tagged TOP3α for 2 days. Proteins were extracted by 0.75% Triton X-100 (supernatant) and proteins left in pellet were extracted by 2% SDS. (B) Co-immunoprecipitation of U2OS cells following treatment as in (A). TOP3α proteins extracted by 0.75% Triton X-100 were pulled down by anti-GFP beads. (C) qPCR analysis of mtDNA bound to TOP3α following co-immunoprecipitation as in (B). Results were mean ± SD, $n = 3$ technical replicates. Unpaired $t$-test was used for statistical analysis. (D) Immunoblotting of U2OS cells following transfection of empty vector or corresponding plasmids encoding FLAG-tagged TOP3α constructs for 2 days. Cytosolic fraction was extracted by 0.025% digitonin and the whole-cell lysate was extracted by 2% SDS. (E) Co-immunoprecipitation of U2OS cells following treatment as in (D). Cytosolic TOP3α proteins extracted by 0.025% digitonin were pulled down by anti-FLAG beads. (F) qPCR analysis of mtDNA bound to cytosolic TOP3α following co-immunoprecipitation as in (E). Results were mean ± SD, $n = 3$ technical replicates. Unpaired $t$-test was used for statistical analysis. (G) The biotin-labeled DNA was transfected into U2OS cells for pull down of cytosolic components by streptavidin beads as in Fig. 6A. U2OS cells were transfected with empty vector or corresponding plasmids encoding FLAG-tagged TOP3α constructs for 2 days. (H) U2OS cells expressing EGFP or EGFP tagged TOP3α for 2 days and subjected to interaction analysis as in Fig. 6C.

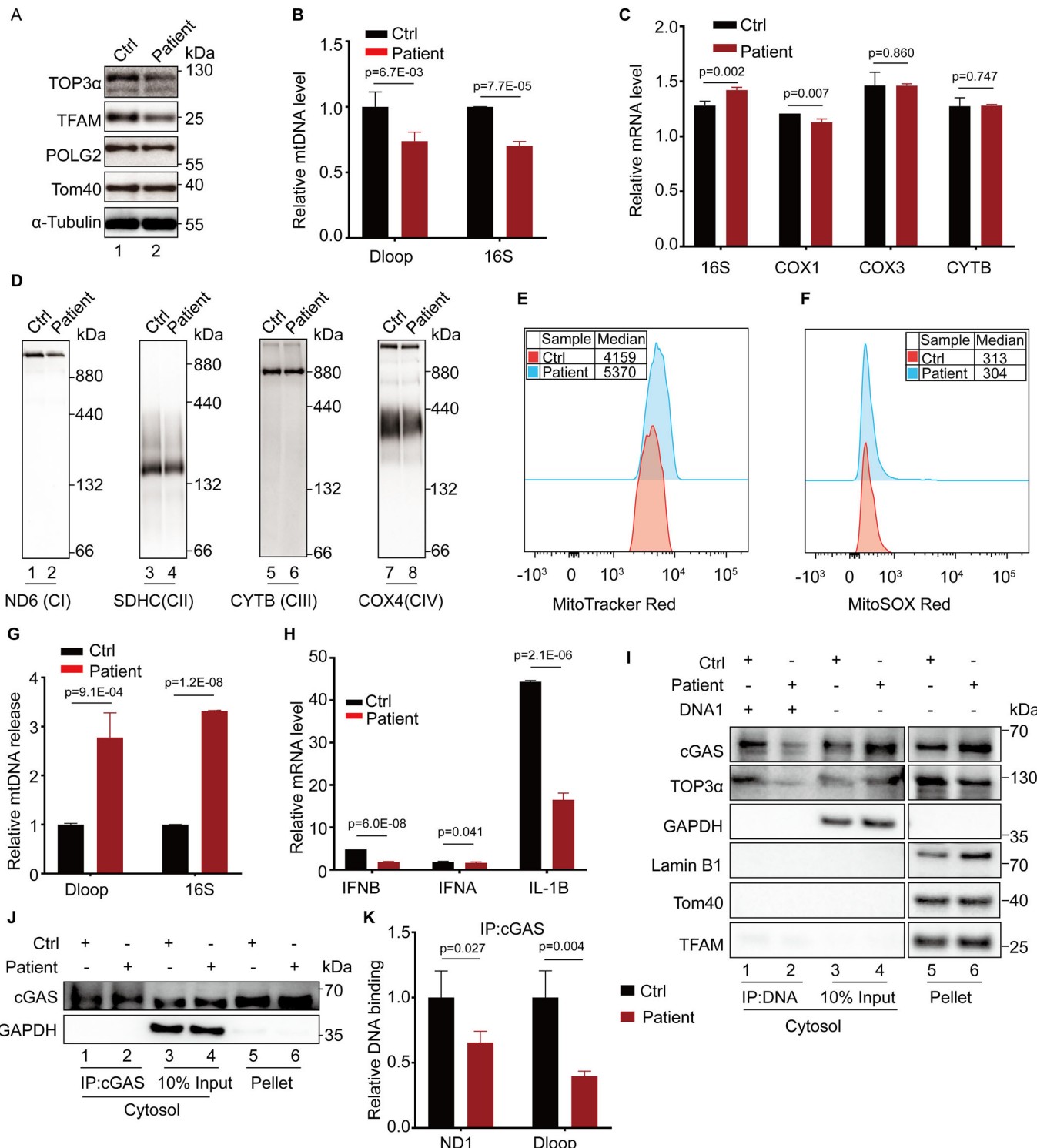

◀ **Figure EV5.  G250D mutation impairs mtDNA interaction with cytosolic TOP3α and cGAS in patient-derived PBMCs.**

(A) Immunoblotting of peripheral blood mononuclear cells (PBMCs) isolated from ALS patient and a control in this family. (B) qPCR for mtDNA copy number in PBMCs. Results were mean ± SD, $n = 3$ technical replicates. Unpaired *t*-test was used for statistical analysis. (C) qPCR for mRNA level in PBMCs. Results were mean ± SD, $n = 3$ technical replicates. Unpaired *t*-test was used for statistical analysis. (D) Immunoblotting of protein complexes extracted by 1% digitonin of PBMCs. (E) Flow cytometry of PBMCs, cells were stained with 100 nM MitoTracker Red for 30 min. (F) Flow cytometry of PBMCs, cells were stained with 5 μM MitoSOX Red for 30 min. (G) qPCR for mtDNA release in PBMCs. Results were mean ± SD, $n = 3$ technical replicates. Unpaired *t*-test was used for statistical analysis. (H) qPCR for mRNA level in PBMCs. Results were mean ± SD, $n = 3$ technical replicates. Unpaired *t*-test was used for statistical analysis. (I) Cytosolic components of PBMCs were extracted using 0.025% digitonin and subjected to immunoprecipitation by biotin-labeled DNA. The pellets were lysed with 2% SDS. (J) Co-immunoprecipitation of cytosolic components via anti-cGAS beads. Cytosolic and pellet components were extracted as described in (I). (K) qPCR analysis of mtDNA bound to endogenous cGAS following co-immunoprecipitation as in (J). Results were mean ± SD, $n = 3$ technical replicates. Unpaired *t*-test was used for statistical analysis.

