## [Peer Review File · EMBO Reports]

Cytosolic TOP3 α facilitates mitochondrial DNA sensing by cGAS

Dongjing Cai, Cheng Chen, Piyanat Meekrathok, Weiqian Zeng, Zheng Wang, Zhigang Peng, Yunan Mo, Xia Xu, Junling Wang, and Jian Qiu

Corresponding author(s): Jian Qiu (qiuqian@sklmg.edu.cn)

Review Timeline:

Submission Date:	5th May 25
Editorial Decision:	28th May 25
Revision Received:	21st Aug 25
Editorial Decision:	30th Sep 25
Revision Received:	12th Oct 25
Accepted:	14th Oct 25

Editor: Esther Schnapp

Transaction Report:

Dear Prof. Qiu,

Thank you for the submission of your manuscript to EMBO reports. We have now received the full set of referee reports that is pasted below.

As you will see, the referees acknowledge that the findings are potentially interesting. However, they also all have several concerns and suggestions for how the study and data should be strengthened and improved. I think all suggestions are good and should be addressed, except points 6 from referee 2 and referee 3, which do not necessarily need to be addressed experimentally. Please let me know if you disagree, and we can discuss the exact revision requirements further, also in a video chat, if you like.

I would thus like to invite you to revise your manuscript with the understanding that the referee concerns must be fully addressed and their suggestions taken on board. Please address all referee concerns in a complete point-by-point response. Acceptance of the manuscript will depend on a positive outcome of a second round of review. It is EMBO reports policy to allow a single round of major revision only and acceptance or rejection of the manuscript will therefore depend on the completeness of your responses included in the next, final version of the manuscript.

We realize that it is difficult to revise to a specific deadline. In the interest of protecting the conceptual advance provided by the work, we recommend a revision within 3 months (28th Aug 2025). Please discuss the revision progress ahead of this time with the editor if you require more time to complete the revisions.

- 1) A data availability section providing access to data deposited in public databases is missing. If you have not deposited any data, please add a sentence to the data availability section that explains that.
- 2) Your manuscript contains statistics and error bars based on $n=2$. Please use scatter blots in these cases. No statistics should be calculated if $n=2$.

3) We replaced Supplementary Information with Expanded View (EV) Figures and Tables that are collapsible/expandable online. A maximum of 5 EV Figures can be typeset. EV Figures should be cited as 'Figure EV1, Figure EV2' etc... in the text and their respective legends should be included in the main text after the legends of regular figures.

5) a complete author checklist, which you can download from our author guidelines <https://www.embopress.org/page/journal/14693178/authorguide>. Please insert information in the checklist that is also reflected in the manuscript. The completed author checklist will also be part of the RPF.

6) Please note that all corresponding authors are required to supply an ORCID ID for their name upon submission of a revised manuscript (<https://orcid.org/>). Please find instructions on how to link your ORCID ID to your account in our manuscript tracking system in our Author guidelines <https://www.embopress.org/page/journal/14693178/authorguide#authorshipguidelines>

12) All Materials and Methods need to be described in the main text using our 'Structured Methods' format, which is required for all research articles. According to this format, the Methods section includes a Reagents and Tools Table (listing key reagents, experimental models, software and relevant equipment and including their sources and relevant identifiers) followed by a Methods and Protocols section describing the methods using a step-by-step protocol format. The aim is to facilitate adoption of the methodologies across labs. More information on how to adhere to this format as well as a downloadable template (.docx) for the Reagents and Tools Table can be found in our author guidelines: <https://www.embopress.org/page/journal/14693178/authorguide#structuredmethods>.

An example of a Method paper with Structured Methods can be found here: <https://www.embopress.org/doi/full/10.1038/s44320-024-00037-6#sec-4>

I look forward to seeing a revised form of your manuscript when it is ready.

Referee #1:

Overall response:

The manuscript entitled 'Cytosolic TOP3alpha modulates mitochondrial DNA sensing by cGAS' by Cai et al seeks to determine the function of a previously under studied, small pool of cytosolic TOP3alpha. In doing so, the authors have uncovered a role for this cytosolic TOP3alpha in assisting the cytoplasmic DNA sensor cGAS in detecting mtDNA that has been released from the mitochondria, and ultimately modulate the strength of the inflammatory response following mtDNA release. Moreover, the authors identify a point mutation in TOP3alpha that impact cGAS sensing of mtDNA release. Their findings are novel, and interesting to the field providing novel insights into mtDNA release and inflammatory signalling, as well as furthering our understanding of how the detection of mtDNA by cGAS occurs. There are a number of concerns that should be addressed:

Major comments:

1. While the authors demonstrated the existence of the cytoplasmic pool of TOP3alpha by sub-cellular fractionation, more evidence is required to demonstrate that TOP3alpha is cytoplasmic within living cells, and that this is not an artefact of the fractionation method, particularly given that it's such a small pool that's detected. Might the TOP3alpha that is detected be coming out of mitochondrial pores with the mtDNA? This would be similar to TFAM.
2. Furthermore, short of a small sentence in the discussion stating that 'cytosolic TOP3alpha is determined by the efficiency of mitochondrial and nuclear import', no evidence is provided as to why the cytosolic TOP3alpha is not imported, despite presence of a mitochondrial targeting sequence. Is the cytoplasmic form a different isoform, which is why it isn't imported?
3. Throughout, the authors use what is effectively an overexpression model of cytoplasmic TOP3alpha through the use of the -MTS-TOP3alpha construct to demonstrate that modulating TOP3alpha levels influences DNA sensing. Can the cytoplasmic form of TOP3alpha be depleted?
4. While the authors show an initial sub cell fractionation, indicating that their conditions can isolate a cytoplasmic fraction, I'd like to see positive controls, indicating a clean fractionation for each of the IPs (e.g. Fig 3 .C, H Fig 4. B). This was done for Fig 3.D, and it would be appropriate for the other IPs.
5. In figure 4, the authors claim that TOP3 α directly facilitates the interaction of mtDNA with cGAS. While the DNA IP blot does potentially suggest this, I don't think this is sufficient evidence on its own. Other measures to directly measure the kinetics of the cGAS-DNA interaction in the presence of titrated TOP3alpha would strengthen the findings.
6. Fig. 4G - Might the C-terminal domain affect the activity of the protein? The authors should test this in cell expression studies. In the pulldowns, they should perform no DNA controls to ensure that the heating and cooling of TOP3alpha does not simply lead to aggregation and non-specific association with the beads. More careful controls are needed here.
7. When the WT TOP3alpha is re-expressed in the siTOP3alpha treated backgrounds, I would have expected a reduction in mtDNA release and inflammatory mRNA, given that loss of TOP3alpha promoted mtDNA release and inflammatory mRNA production? Can the authors comment on why that isn't the case?

Other comments:

- Fig 2:
8. Panel A was done on control cells but not TOP3A depleted cells. This should be done to verify that mitochondrial membranes are not affected by changes in mtDNA structure
 9. While CSA and VBIT-4 did reduce mRNA levels of most inflammatory genes back to control levels, IFNA is still noticeably increased relative to control. Can the authors comment on this?
 10. Panel G - cGAS knockdown is quite inefficient and surprising that the effects are quite marked (2H) given that cGAS is an enzyme. It's even lower than the control which surprises me. Can the authors explain?

11. siRNA experiments should be complemented with rescues.

Fig 5:

12. panel H: Is the TOP3 α FLAG-tagged here? It's unclear which construct possesses the FLAG tag in the figure or figure legend, please clarify.

13. panel J: Can the authors include a loading control for the BN-PAGE?

Referee #2:

This study reveals a previously undefined role for cytosolic TOP3 α in amplifying mitochondrial DNA (mtDNA)-triggered innate immune responses. The authors demonstrated that aberrant TOP3 α expression induces mtDNA clustering and release via the mPTP-VDAC axis, activating cGAS-mediated inflammation. Cytosolic TOP3 α enhances cGAS sensing of mtDNA and amplifies downstream signaling. The study is interesting, and most the conclusions are supported by the data.

However, there are still some problems in this paper, and more experiments are needed to improve this study. The specific opinions are as follows:

1. The observation that both top3 deletion and overexpression enhance mtDNA release is notably paradoxical. This needs reasonable explanation. Further evaluation of the effects of top3 deletion/overexpression on mitochondrial function is warranted. More experiments are needed to explain the effect of top3 on mtDNA release.
2. FigureS3C showed that there was no significant change in the mitochondrial respiratory chain complex after TOP3 α deletion. However, in the article "Two type I topoisomerases maintain DNA topology in human mitochondria", the author found that the deletion of TOP3 α reduced complex III. Please explain the difference in this result.
3. In Figure 3H, co-immunoprecipitation experiments were conducted on the cytoplasmic components of HMC3 cells expressing EGFP or EGFP-labeled TOP3 α constructs using anti-cGAS beads. How can it be proved that the expression of the mts deletion version of TOP3 α promotes the binding of cytoplasmic mtDNA to cGAS? The experimental design was wrong.
4. It is recommended to repeat the experiments in U2OS cells using immune cells (for example: Figure1, Figure3 C, Figure5 D-H, Figure6 B,D).
5. It is recommended to employ cGAS-knockout cells to further elucidate the regulatory mechanism of TOP3 α and validate the specificity of the cGAS-associated targeting pathway.
6. The article should elaborate on the clinical diseases caused by TOP3 α mutations in the introduction and establish therapeutic animal models to validate potential treatment strategies.

Referee #3:

Mitochondria are essential organelles whose dysfunction is linked to numerous disorders. In the context of innate immunity, mitochondrial integrity is crucial to prevent the leakage of mitochondrial DNA (mtDNA) into the cytoplasm, which leads to recognition by cyclic GMP-AMP synthase (cGAS), activation of STING, and subsequent (aberrant) induction of interferons. Topoisomerase 3 α (TOP3 α) belongs to the type IA topoisomerase subfamily and shows dual subcellular localization in both mitochondria and the nucleus. In mitochondria, TOP3 α plays an important role in the decatenation and segregation of mtDNA. Intriguingly, a minor fraction of TOP3 α has also been observed in the cytosol, but its functional relevance remains completely unknown.

Here, Cai et al. propose that cytosolic TOP3 α facilitates the sensing of released mtDNA by cGAS, enhances the interaction between cGAS and mtDNA, and amplifies downstream innate immune signaling. Additionally, they suggest that TOP3 α competes with cGAS for mtDNA binding. A rare, patient-associated mutation in TOP3 α (G250D) is proposed to impair cGAS-mtDNA interaction, thereby compromising cGAS-mediated immune activation.

While the study presents interesting findings, key controls are missing to support the main conclusions. Additionally, experiments in more physiologically relevant systems (e.g., primary cells) are lacking. Finally, it remains unclear whether the proposed mechanism for G250D variant is relevant for the patient phenotype, raising questions about the broader significance of the findings.

Major Issues

1. Important immunoblots are missing. Include blots for TOP3 α (Fig. 5D,G) and GAPDH (Fig. 3C, 6B,D). Similarly, it is essential to stain for the immunoprecipitated protein to ensure similar IP efficiency, which is crucial for drawing valid conclusions. This includes Fig. 6E (stain for cGAS) and Fig. 3C (stain for biotinylated DNA).
2. Fig. 3F-I: It is possible that Δ Mts increases mtDNA release. This would indicate that increased ISG mRNA levels (Fig. 3G) and increased DNA binding (Fig. 3I) upon Δ Mts expression are indirect effects caused by increased mtDNA release. Please quantify the levels of cytosolic mtDNA upon WT and Δ Mts expression. If there is more cytosolic mtDNA in the presence of Δ Mts, all conclusions would probably be invalid.
3. The conclusions drawn from Fig. 6B,D are not apparent from the blots shown. Please quantify at least three blots from independent experiments to validate the conclusions. Additionally, improve the cGAS staining for the input in Fig. 6B.
4. Similarly, for Fig. 3C, quantify blots from three independent experiments.

5. Repeat key experiments in primary cells. Please confirm the mechanism of G250D (reduced mtDNA binding and reduced binding of cGAS to mtDNA) in primary (patient) cells.
6. Challenge G250D-expressing cells with DNA viruses (e.g., HSV-1) and analyze the immune response. Does TOP3 α also affect cGAS binding to non-mitochondrial DNA, such as virus-derived DNA?

Minor Issues

1. Fig. 3J,K: The expression levels of TOP3 α WT and TOP3 α Δ Mts are very low and differ substantially, which could affect the experimental outcome. Please repeat the experiments with more similar expression levels.
2. Please arrange the figures in a more chronological manner. Sometimes figures are not positioned where expected, making it more difficult for the reader (e.g., Fig. 5, 6).
3. Line 170: siTOP3 α treatment depletes TOP3 α not specifically in the cytosol but throughout the cell. Please remove this statement.
4. Provide more details in the figure legends and specify the nature of the replicates (technical, biological, etc.).

Referee #1:

Overall response:

The manuscript entitled 'Cytosolic TOP3alpha modulates mitochondrial DNA sensing by cGAS' by Cai et al seeks to determine the function of a previously under studied, small pool of cytosolic TOP3alpha. In doing so, the authors have uncovered a role for this cytosolic TOP3alpha in assisting the cytoplasmic DNA sensor cGAS in detecting mtDNA that has been released from the mitochondria, and ultimately modulate the strength of the inflammatory response following mtDNA release. Moreover, the authors identify a point mutation in TOP3alpha that impact cGAS sensing of mtDNA release. Their findings are novel, and interesting to the field providing novel insights into mtDNA release and inflammatory signalling, as well as furthering our understanding of how the detection of mtDNA by cGAS occurs. There are a number of concerns that should be addressed:

Response: Thank you very much for your kind words regarding the novelty and importance of our work as well as for the important questions you raised.

Major comments:

1. While the authors demonstrated the existence of the cytoplasmic pool of TOP3alpha by sub-cellular fractionation, more evidence is required to demonstrate that TOP3alpha is cytoplasmic within living cells, and that this is not an artefact of the fractionation method, particularly given that it's such a small pool that's detected. Might the TOP3alpha that is detected be coming out of mitochondrial pores with the mtDNA? This would be similar to TFAM.

Response: Thank you for your important question. We used two fractionation methods both of which are used in numerous papers. For example, digitonin-based method was used in PINK1 study (PMID: 25609704) and mtDNA release study (PMID: 31857488). In our practice, the low concentration of digitonin could extract cytosolic components like α -Tubulin, but not mitochondrial matrix components including TFAM (see Figure 2A), GRP75 and POLG2 (see Figure EV4D). Moreover, the same conclusion applies to PBMCs (Appendix Figure S2A). Similarly, homogenization method could extract cytosolic components like α -Tubulin, but not TFAM from mitochondrial matrix (Appendix Figure S2B). Therefore, these data convinced us that the cytosolic TOP3 α we observed was not due to an artefact of the fractionation method.

2. Furthermore, short of a small sentence in the discussion stating that 'cytosolic TOP3alpha is determined by the efficiency of mitochondrial and nuclear import', no evidence is provided as to why the cytosolic TOP3alpha is not imported, despite presence of a mitochondrial targeting sequence. Is the cytoplasmic form a different isoform, which is why it isn't imported?

Response: Thank you for raising up this important question. While we have no evidence to support that cytosolic TOP3 α is a different isoform or a different post-translationally modified pool of proteins, we think that the subcellular localization of TOP3 α is under intricate regulation. As summarized in Figure EV3B, immunofluorescent

analysis revealed, in the same pool of cells, different patterns of TOP3 α distribution (nuclear enriched, mitochondrial enriched or both) which apparently could affect mtDNA aggregation. The molecular mechanisms regulating the subcellular localization of TOP3 α are under active investigation by other members in our lab. We hope to report the findings to the scientific community in the future.

3. Throughout, the authors use what is effectively an overexpression model of cytoplasmic TOP3alpha through the use of the -MTS-TOP3alpha construct to demonstrate that modulating TOP3alpha levels influences DNA sensing. Can the cytoplasmic form of TOP3alpha be depleted?

Response: Inspired by your important question, we tried to deplete cytoplasmic TOP3 α by enhancing its mitochondrial import. To this end, we fused tandem MTSs (4xMTS) derived from human COX8A to the N-terminus of TOP3 α (Appendix Figure S6A). As expected, this enhanced the mitochondrial targeting of TOP3 α , but unfortunately caused the depletion of mitochondrial membrane potential (Appendix Figure S6B-C), preventing further analysis of such construct. Therefore, we have not found a practical way to specifically deplete cytoplasmic TOP3 α .

4. While the authors show an initial sub cell fractionation, indicating that their conditions can isolate a cytoplasmic fraction, I'd like to see positive controls, indicating a clean fractionation for each of the IPs (e.g. Fig 3 .C, H Fig 4. B). This was done for Fig 3.D, and it would be appropriate for the other IPs.

Response: Thank you for your kind request. You are absolutely right. We think that this concern may be related to your point 1. As we have mentioned in the response to point 1, in our practice, the published fractionation methods work reliably across different cell lines in multiple experiments (Figure 2A, Figure EV4D, Appendix Figure S2A, Appendix Figure S2B), demonstrating that cytosolic components, instead of mitochondrial matrix components, could be extracted. Therefore, we did not routinely check the extraction efficiency and specificity in every IPs.

Since Referee #2 requested to repeat Figure 3C in immune cells (his/her point 4), following the kind requests of you two, we repeated the experiment of Figure 3C in HMC3 cells (as shown in Appendix Figure S3 with GAPDH as control). For the extraction efficiency and specificity in cGAS IP experiments (like Figure 3H), please refer to Figure 6E in the revised manuscript (GAPDH and TFAM as control) and Figure EV5J (GAPDH as control) as examples. As for Figure 4B you mentioned, we guess you meant Figure 6B (now Figure EV4G). Accordingly, we also repeated the experiments in HMC3 cells with proper control and replaced Figure 6B and 6D with the new data.

5. In figure 4, the authors claim that TOP3 α directly facilitates the interaction of mtDNA with cGAS. While the DNA IP blot does potentially suggest this, I don't think this is sufficient evidence on its own. Other measures to directly measure the kinetics of the cGAS-DNA interaction in the presence of titrated TOP3alpha would strengthen the findings.

Response: We appreciate your thoughtful suggestion. Since the purified TOP3 α enhanced the interaction of purified mtDNA fragments with purified cGAS in a biochemical system, we think that our conclusion is valid (Figure 4G, compare lane 5 to lane 10). Moreover, this effect of TOP3 α is concentration dependent (Figure 4G, compare lane 5 to lane 6) and heat sensitive (Figure 4G, compare lane 5 to lane 7). Therefore, although we are not a biophysical lab to measure the precise kinetics of cGAS-DNA interaction upon titrated TOP3 α , we are convinced that TOP3 α directly facilitates the interaction of mtDNA with cGAS at the given experimental condition.

6. Fig. 4G - Might the C-terminal domain affect the activity of the protein? The authors should test this in cell expression studies. In the pulldowns, they should perform no DNA controls to ensure that the heating and cooling of TOP3 α does not simply lead to aggregation and non-specific association with the beads. More careful controls are needed here.

Response: Thank you for your thoughtful suggestion. The function of TOP3 α C-terminal domain is unknown (containing cryptic nuclear localization signal as reported in PMID: 12209014) and deserves further investigation in the future. With the data presented in Figure 4, we think that the facilitating effect of TOP3 α on mtDNA-cGAS interaction does not require the C-terminal domain at the given experimental condition. Meanwhile, we attempted to purify full length TOP3 α , but did not succeed so far (we managed to purify TOP3 α amino acids 21-750 similar to the region reported in PMID: 24509834).

Since we could not exclude the possibility you pointed out without no DNA control, we deleted the following sentence in the revised manuscript: "Of note, preheating did not affect the binding of TOP3 α to mtDNA, suggesting that the mtDNA interacting domain was properly refolded after cooling and was independent of the functional domain promoting cGAS-DNA interaction." By doing so, it does not affect other conclusions or logic flow related to Figure 4.

7. When the WT TOP3 α is re-expressed in the siTOP3 α treated backgrounds, I would have expected a reduction in mtDNA release and inflammatory mRNA, given that loss of TOP3 α promoted mtDNA release and inflammatory mRNA production? Can the authors comment on why that isn't the case?

Response: Thank you for your incisive question. Indeed, this observation is also contradictory to our expectation. We think that this may be caused by the elevated level of re-expressed TOP3 α , exceeding the tolerance of mitochondria and leading to additional mtDNA-related stress. As shown in Figure EV3A and EV3B, enhanced TOP3 α in mitochondria was correlated with abnormal aggregation of mtDNA. Noticing this limit of current experimental setup, we paid special attention to the expression levels between WT and any given mutant used in this study and confirmed that they are similar to each other. Therefore, we think that our conclusions are valid when comparing WT and any given mutant at such experimental condition. To overcome the shorts of overexpression, in our follow-up project, we will test different promoters driving the expression of TOP3 α (aiming to achieve endogenous level) and, in parallel, make necessary effort to tag *TOP3A* at the endogenous gene locus for further studies (current

TOP3 α antibodies could not recognize endogenous level TOP3 α in immunofluorescence experiment).

Other comments:

Fig 2:

8. Panel A was done on control cells but not TOP3A depleted cells. This should be done to verify that mitochondrial membranes are not affected by changes in mtDNA structure

Response: Thank you for your kind suggestion. As shown in Figure EV1A, following TOP3 α knockdown, mitochondrial morphology (OPA1 signal) and mitochondrial membrane potential (MitoTracker Red signal) showed no significant alteration. Moreover, as shown in Figure 6E of the revised manuscript, depleting TOP3 α did not cause obvious detection of TFAM in the cytosolic fraction (comparing lane 4 to lane 5). Therefore, our data indicated that the changes in mtDNA structure upon TOP3 α depletion have not affected mitochondrial membrane integrity in a general way.

9. While CSA and VBIT-4 did reduce mRNA levels of most inflammatory genes back to control levels, IFNA is still noticeably increased relative to control. Can the authors comment on this?

Response: We appreciate your careful observation. We don't know why IFNA is still noticeably increased relative to control in Figure 2F after CSA or VBIT-4 treatment. We guess it may reflect heterogeneous response to the chemicals at certain level. Since the inhibitory effects of CSA and VBIT-4 on mtDNA-related inflammatory responses have been demonstrated in other reports (e.g. PMID: 35835107 and PMID: 33031745), and also worked well in our hands in different experiments (Figure 2F and Figure EV2C), we did not follow the mechanisms underlying the possible heterogeneous response to the chemicals. Thank you for bringing it up.

10. Panel G - cGAS knockdown is quite inefficient and surprising that the effects are quite marked (2H) given that cGAS is an enzyme. It's even lower than the control which surprises me. Can the authors explain?

Response: Thank you for your kind question. Although cGAS knockdown did not lead to undetectable level of cGAS proteins, it still obviously reduced the cGAS expression in our view. The inflammatory genes were analyzed under basic culturing condition without additional stimulation. We think that the cells require a certain level of cGAS proteins to maintain the steady level of inflammatory gene expression under such experimental condition.

11. siRNA experiments should be complemented with rescues.

Response: Thank you for your kind suggestion. Since the regulatory role of cGAS in inflammatory gene expression is well known, we did not perform rescue in cGAS knockdown experiment. Rescue experiments involving TOP3 α re-expression under

knockdown conditions were performed to evaluate the different performance between TOP3 α WT and mutant (Figure 3F-G, Figure 6E-H).

Fig 5:

12. panel H: Is the TOP3 α FLAG-tagged here? It's unclear which construct possesses the FLAG tag in the figure or figure legend, please clarify.

Response: We are sorry for this confusion. Yes, TOP3 α is FLAG-tagged in Figure 5H (now Figure EV4E). We have explicitly note this in the figure legends in the revised manuscript. Thank you.

13. panel J: Can the authors include a loading control for the BN-PAGE?

Response: Figure 5J is now Figure 5I in the revised manuscript. We attempted to detect cytosolic complexes as loading control via BN-PAGE using various antibodies (including those against LC3, NLRP3, p62, and cGAS). But these antibodies did not perform well under BN-PAGE condition. Therefore, we have not found suitable antibodies for cytosolic complex detection as loading control. However, as shown in Appendix Figure S8, the background signal revealed by NDUFS1 antibodies (lanes 7-9, the same samples loaded as in lanes 1-3) could serve as loading control.

Referee #2:

This study reveals a previously undefined role for cytosolic TOP3 α in amplifying mitochondrial DNA (mtDNA)-triggered innate immune responses. The authors demonstrated that aberrant TOP3 α expression induces mtDNA clustering and release via the mPTP-VDAC axis, activating cGAS-mediated inflammation. Cytosolic TOP3 α enhances cGAS sensing of mtDNA and amplifies downstream signaling. The study is interesting, and most the conclusions are supported by the data.

However, there are still some problems in this paper, and more experiments are needed to improve this study. The specific opinions are as follows:

Response: Thank you very much for your kind words about our work as well as for the important questions you raised.

1. The observation that both top3 deletion and overexpression enhance mtDNA release is notably paradoxical. This needs reasonable explanation. Further evaluation of the effects of top3 deletion/overexpression on mitochondrial function is warranted. More experiments are needed to explain the effect of top3 on mtDNA release.

Response: Thank you for your important comments and suggestions. Our data demonstrated that depleting TOP3 α caused mtDNA aggregation and release (Figure 1) without affecting mitochondrial morphology and membrane potential (Figure EV1A). We also demonstrated that TOP3 α overexpression could lead to mtDNA aggregation (Figure EV3A-B) and release (Figure EV3D). These data reflected that a proper

expression level of TOP3 α is important for mtDNA homeostasis. Too little or too much TOP3 α could both cause abnormal mtDNA aggregation and release.

As shown in Figure EV1A, TOP3 α knockdown did not obviously affect mitochondrial morphology and membrane potential in U2OS cells, indicating that mtDNA release upon TOP3 α depletion was not due to a general impairment of mitochondrial membrane integrity. Following your kind suggestion, similar observations were made in HEK293T cells. TOP3 α knockdown did not cause noticeable alterations in mitochondrial membrane potential (Appendix Figure S1A) and mitochondrial ROS level (Appendix Figure S1B), neither obviously affect mitochondrial protein complexes involved in mtDNA maintenance/expression and OXPHOS (Appendix Figure S1C). As shown in Figure EV3A-C, TOP3 α overexpression could lead to mtDNA aggregation, but did not obviously affect mitochondrial protein complexes involved in mtDNA homeostasis and OXPHOS. At such condition, TOP3 α overexpression did not cause obvious alterations in mitochondrial membrane potential (Figure EV3F) and mitochondrial ROS level (Figure EV3G).

In our view, it seems that mtDNA aggregation caused by abnormal levels of TOP3 α is the key leading to mtDNA fragmentation and release without general impairment of mitochondrial membrane integrity. We reason that the mechanisms underlying mtDNA aggregation observed in the presence of too much or too little TOP3 α may be different. Our favorite way to dissect the underlying mechanisms is to develop a fluorescent reporter which could be activated by cytosolic mtDNA. Combined with genome-wide CRISPR/Cas9 screening at different conditions (including abnormal levels of TOP3 α), novel components regulating mtDNA release may be identified. We hope to achieve this and report the findings to the scientific community in the future.

2. FigureS3C showed that there was no significant change in the mitochondrial respiratory chain complex after TOP3 α deletion. However, in the article "Two type I topoisomerases maintain DNA topology in human mitochondria", the author found that the deletion of TOP3 α reduced complex III. Please explain the difference in this result.

Response: Thank you for your comments. In Figure 5F of the work you pointed out, Menger *et al.* demonstrated that TOP3 α knockdown in HeLa cells reduced the protein level of UQCRC2 (Complex III subunit) without affecting the levels of NDUFA9 (Complex I subunit), SDHA (Complex II subunit), COXIV (Complex IV subunit) and ATPB (Complex V subunit) after SDS-PAGE and immunoblotting analysis. In Figure S3C (now Figure EV3C) of our manuscript, we overexpressed TOP3 α instead of depletion in U2OS cells, and did not observe obvious alterations to Complex III detected by UQCRC1 antibodies after BN-PAGE and immunoblotting analysis. Thus, we overexpressed TOP3 α , instead of depletion, in different type of cells, and analyzed the sample at complex level (by BN-PAGE) instead of subunit level (by SDS-PAGE).

3. In Figure 3H, co-immunoprecipitation experiments were conducted on the cytoplasmic components of HMC3 cells expressing EGFP or EGFP-labeled TOP3 α constructs using anti-cGAS beads. How can it be proved that the expression of the mts deletion version of TOP3 α promotes the binding of cytoplasmic mtDNA to cGAS? The experimental design was wrong.

Response: Thank you for your important comment. In Figure 3H, we demonstrated that similar amounts of cGAS proteins were enriched by the beads. Subsequently, the enriched material was subjected to qPCR analysis for the bound mtDNA (as shown in Figure 3I). We observed elevated signal of mtDNA in the sample expressing the MTS-deleted version of TOP3 α comparing to the sample expressing WT. Therefore, we are convinced that the expression of MTS-deleted version of TOP3 α promoted the binding of cytosolic mtDNA to cGAS.

4. It is recommended to repeat the experiments in U2OS cells using immune cells (for example: Figure1, Figure3 C, Figure5 D-H, Figure6 B,D).

Response: Thank you for your important suggestion. The cytoplasmic space of HMC3 and BV2 cells is quite limited and not suitable for imaging of mtDNA release. We therefore used U2OS cells in Figure 1. To analyze mtDNA release in BV2 and HMC3 cells, we used qPCR analysis as shown in Figure 2C and Figure 2E.

For Figure 3C, we performed further experiment using HMC3 cells following your kind suggestion. As shown in Appendix Figure S3, depleting TOP3 α impaired the interaction of cytosolic cGAS with mtDNA-derived fragments, consistent with our observation in U2OS cells. Moreover, TOP3 α could also promote the interaction of cGAS with GFP-derived sequence, indicating that the effect of cytosolic TOP3 α on cGAS is not limited to self-DNA. This may have important implications under virus-infected conditions and deserves further investigation in the future.

For Figure 5D-I (now Figure EV4A-F), we also performed further experiments using HMC3 cells following your kind suggestion. As shown in Figure 5D-F of the revised manuscript, G250D mutation did not affect the protein level of mitochondrial TOP3 α , but impaired its interaction with mtDNA. Moreover, G250D mutation diminished the binding of mtDNA to cytosolic TOP3 α without affecting the protein steady level (Figure 5G-H of the revised manuscript). These data are consistent with our observation in U2OS cells (Figure EV4A-F). For Figure 6B and 6D (now Figure EV4G-H), we also obtained similar observation in HMC3 cells and replaced them with the new data.

5. It is recommended to employ cGAS-knockout cells to further elucidate the regulatory mechanism of TOP3 α and validate the specificity of the cGAS-associated targeting pathway.

Response: Thank you for your kind suggestion. The role of cGAS as a primary cytosolic sensor for mtDNA is well-established. Since our major finding is that cytosolic TOP3 α promotes mtDNA sensing by cGAS, to further dissect the molecular mechanisms how TOP3 α achieves this, we feel making cGAS-knockout cells is not our immediate urgent task. Instead, we are trying to elucidate the structure of TOP3 α -mtDNA-cGAS complex and hope to report the findings to the scientific community in the future.

6. The article should elaborate on the clinical diseases caused by TOP3 α mutations in the introduction and establish therapeutic animal models to validate potential treatment strategies.

Response: Thank you for your kind suggestion. Regarding the clinical diseases caused by TOP3 α mutations, we have rephrased the sentence “The pathogenic mutations of TOP3A, which encodes TOP3 α , have been associated with numerous diseases with a broad clinical spectrum” as follows: “The pathogenic mutations of TOP3 α (such as M100V, A176V and D479G) have been associated with numerous diseases (including mitochondrial diseases and Bloom syndrome) with a broad clinical spectrum”.

We think that it is interesting and important to establish TOP3 α -related disease animal models and search for potential treatment strategies. However, we regret that currently we do not have sufficient funding and human power to do so. We hope that we could implement this in the future and thank you for bring it up.

Referee #3:

Mitochondria are essential organelles whose dysfunction is linked to numerous disorders. In the context of innate immunity, mitochondrial integrity is crucial to prevent the leakage of mitochondrial DNA (mtDNA) into the cytoplasm, which leads to recognition by cyclic GMP-AMP synthase (cGAS), activation of STING, and subsequent (aberrant) induction of interferons. Topoisomerase 3 α (TOP3 α) belongs to the type IA topoisomerase subfamily and shows dual subcellular localization in both mitochondria and the nucleus. In mitochondria, TOP3 α plays an important role in the decatenation and segregation of mtDNA. Intriguingly, a minor fraction of TOP3 α has also been observed in the cytosol, but its functional relevance remains completely unknown.

Here, Cai et al. propose that cytosolic TOP3 α facilitates the sensing of released mtDNA by cGAS, enhances the interaction between cGAS and mtDNA, and amplifies downstream innate immune signaling. Additionally, they suggest that TOP3 α competes with cGAS for mtDNA binding. A rare, patient-associated mutation in TOP3 α (G250D) is proposed to impair cGAS-mtDNA interaction, thereby compromising cGAS-mediated immune activation.

While the study presents interesting findings, key controls are missing to support the main conclusions. Additionally, experiments in more physiologically relevant systems (e.g., primary cells) are lacking. Finally, it remains unclear whether the proposed mechanism for G250D variant is relevant for the patient phenotype, raising questions about the broader significance of the findings.

Response: Thank you very much for your important comments. We have performed additional experiments to fulfill your kind suggestions/requests.

Major Issues

1. Important immunoblots are missing. Include blots for TOP3 α (Fig. 5D,G) and GAPDH (Fig. 3C, 6B,D). Similarly, it is essential to stain for the immunoprecipitated protein to ensure similar IP efficiency, which is crucial for drawing valid conclusions. This includes Fig. 6E (stain for cGAS) and Fig. 3C (stain for biotinylated DNA).

Response: Thank you for your important comments. Figures 5D and 5G (now Figures EV4A and EV4D) serve as controls validating the efficiency of subcellular fractionation, with the corresponding TOP3 α detection shown in Figures 5E and 5H (now Figures EV4B and EV4E), respectively. Here we present the complete blots of Figure EV4B. As shown in Appendix Figure S7, the extraction efficiency of TOP3 α is lower than TFAM (Figure EV4A, the same sample as in Figure EV4B) possibly due to the nuclear presence of TOP3 α . In our practice, TFAM is a better marker for mitochondrial extraction at this condition (0.75% Triton X-100). For Figure 5G (now Figure EV4D), for similar reason, we usually use abundant cytosolic proteins (such as α -Tubulin) as control for cytosolic extraction by 0.025% digitonin.

For Figures 3C, 6B and 6D, we are sorry that we did not include GAPDH as control. We repeated these experiments in HMC3 cells and included GAPDH as control. For Figure 3C, as shown in Appendix Figure S3, depleting TOP3 α impaired the interaction of cytosolic cGAS with mtDNA-derived fragments, consistent with our observation in U2OS cells. GAPDH could not be pulled down by DNA. For Figures 6B and 6D (now Figure EV4G-H), we also obtained similar observation in HMC3 cells, and again GAPDH could not be pulled down by DNA and we replaced Figure 6B and 6D with the new data.

For Figure 6E (now removed in the revised manuscript because of redundancy), we repeated the experiment in HMC3 cells and included cGAS as control. Similar IP efficiency was obtained for cGAS in different samples (new Figure 6E), with less mtDNA bound to cGAS in the sample expressing TOP3 α G250D mutant comparing to the WT (new Figure 6F). This confirmed our previous observation.

As for the IP efficiency of biotinylated DNA, after DNA pull-down, we boiled the streptavidin beads with SDS and analyzed the DNA by agarose gel for GelRed signal. As shown in Appendix Figure S10, the IP efficiency of biotinylated DNA was similar in different samples. Moreover, in IP DNA experiments, the presence of more TOP3 α promoted DNA-cGAS interaction in Figure 3C, Figure 4G (compare lane 5 to lane 10), Figure EV4G (compare lane 1 to lane 2), Figure EV4H (compare lane 1 to lane 2), Appendix Figure S3 (compare lane 1 to lane 2, and lane 3 to lane 4), new Figure 6B (compare lane 1 to lane 2) and new Figure 6D (compare lane 1 to lane 2). This conclusion is further supported by IP cGAS experiments (Figure 3I EGFP vs WT, new Figure 6E-F EGFP vs WT). Taken together, we are convinced that our conclusion is unlikely to be caused by different IP efficiency of biotinylated DNA.

2. Fig. 3F-I: It is possible that Δ Mts increases mtDNA release. This would indicate that increased ISG mRNA levels (Fig. 3G) and increased DNA binding (Fig. 3I) upon Δ Mts expression are indirect effects caused by increased mtDNA release. Please quantify the levels of cytosolic mtDNA upon WT and Δ Mts expression. If there is more cytosolic mtDNA in the presence of Δ Mts, all conclusions would probably be invalid.

Response: Thank you for your important suggestion. We quantified the levels of cytosolic mtDNA upon TOP3 α WT and Δ MTS expression. We observed less cytosolic mtDNA in the sample expressing Δ MTS comparing to WT (Appendix Figure S5). As shown in Figure EV3A and EV3B, enhanced TOP3 α in mitochondria was correlated with abnormal aggregation of mtDNA. Together, these data indicated that mitochondrial

overloaded TOP3 α is required to cause mtDNA release. Therefore, increased ISG mRNA levels (Figure 3G) and increased DNA binding (Figure 3I) upon TOP3 α Δ MTS expression was not likely due to increased mtDNA release.

3. The conclusions drawn from Fig. 6B,D are not apparent from the blots shown. Please quantify at least three blots from independent experiments to validate the conclusions. Additionally, improve the cGAS staining for the input in Fig. 6B.

Response: Since Referee #2 requested to repeat Figures 6B and 6D (now Figure EV4G-H in the revised manuscript) in immune cells (his/her point 4), following the kind requests of you two, we repeated the experiment of Figure 6B in HMC3 cells, and quantified the blots. As shown in Appendix Figure S9, G250D mutation indeed impaired the interaction of TOP3 α as well as cGAS to mtDNA. We also repeated the experiment of Figure 6D in HMC3 cells, and quantified the blots. As shown in Appendix Figure S11, G250D mutation again impaired the interaction of TOP3 α as well as cGAS to mtDNA. Figures 6B and 6D were replaced with new data.

4. Similarly, for Fig. 3C, quantify blots from three independent experiments.

Response: Since Referee #2 requested to repeat Figure 3C in immune cells (his/her point 4), following the kind requests of you two, we repeated the experiment of Figure 3C in HMC3 cells, and quantified the blots. As shown in Appendix Figure S4, depleting cytosolic TOP3 α impaired the interaction of cGAS to mtDNA.

5. Repeat key experiments in primary cells. Please confirm the mechanism of G250D (reduced mtDNA binding and reduced binding of cGAS to mtDNA) in primary (patient) cells.

Response: Thank you for your important suggestion. We performed immunoprecipitation experiments in patient-derived PBMCs and observed impaired enrichment of TOP3 α and cGAS by mtDNA fragments (Figure EV5I). Moreover, qPCR analysis demonstrated that patient-derived cells expressing TOP3 α -G250D had reduced amount of mtDNA bound to endogenous cGAS following co-immunoprecipitation of cytosolic components via anti-cGAS beads (Figure EV5J-K).

6. Challenge G250D-expressing cells with DNA viruses (e.g., HSV-1) and analyze the immune response. Does TOP3 α also affect cGAS binding to non-mitochondrial DNA, such as virus-derived DNA?

Response: Thank you for your interesting and important question. Since we are not a lab specialized in virology, we opted to use biotin-labeled GFP-derived DNA (with size and GC content comparable to biotin-labeled mtDNA) to answer this question. As shown in Appendix Figure S3, depleting TOP3 α impaired the interaction of cytosolic cGAS with mtDNA-derived fragments as well as GFP-derived DNA. This interesting observation indicated that the effect of cytosolic TOP3 α on cGAS is not limited to mtDNA. This may

have important implications under pathogen-infected conditions and deserves further investigation in the future.

Minor Issues

1. Fig. 3J,K: The expression levels of TOP3 α WT and TOP3 α Δ Mts are very low and differ substantially, which could affect the experimental outcome. Please repeat the experiments with more similar expression levels.

Response: Thank you for your kind suggestion. In Figure 3J-K, the cytosolic pool of TOP3 α was analyzed, which indeed represents the minor pool of total TOP3 α (also because we have to adapt the exposure time to the strong GFP signal). Δ MTS abolished the mitochondrial localization of TOP3 α , leading to increased signal in the cytosol (without obvious change in total TOP3 α level comparing to WT, also see Figure 3F and Appendix Figure S2B). However, the level of mtDNA bound to the cytosolic TOP3 α Δ MTS was still lower than that of WT (despite of more cytosolic TOP3 α Δ MTS).

2. Please arrange the figures in a more chronological manner. Sometimes figures are not positioned where expected, making it more difficult for the reader (e.g., Fig. 5, 6).

Response: Thank you for your kind suggestion. We have improved the figure arrangement to make it easier for the reader.

3. Line 170: siTOP3 α treatment depletes TOP3 α not specifically in the cytosol but throughout the cell. Please remove this statement.

Response: Thank you for your kind suggestion. Around line 170, we guess you are referring to the following text in lines 175-176: "Surprisingly, depletion of cytosolic TOP3 α impaired the interaction of cGAS to mtDNA without affecting the total level of cGAS in the input (Figure 3C)." To avoid confusion, we rewrote the sentence as: "Surprisingly, depletion of TOP3 α impaired the interaction of cGAS to mtDNA without affecting the total level of cGAS in the input (Figure 3C)."

4. Provide more details in the figure legends and specify the nature of the replicates (technical, biological, etc.).

Response: We are grateful for your kind suggestion. We have indicated technical or biological replicates in the figure legends as you suggested.

Dear Prof. Qiu,

Thank you for the submission of your revised manuscript. We have now received the enclosed reports from the referees, and I am happy to say that all support its publication now. Please address or comment on referee 1's concern. Providing exclusively technical replicates for qPCR is indeed not sufficient.

A few editorial requests will also need to be addressed before we can proceed with the official acceptance of your manuscript.

- Your ms does not contain a "Data Availability Section" (DAS) that needs to be placed before the Acknowledgments. If you have not generated data in this study that is deposited in public databases please mention this fact in the DAS.
- The conflict of interest subheading needs to be renamed to "Disclosure and Competing Interests Statement"
- The REFERENCE format is not correct: et al needs to be used after 10 author names. Please use the EMBO reports reference style.
- The FUNDING INFO is OK, but it needs to go under Acknowledgments and the "Funding" section heading is not needed.
- Please upload the APPENDIX FILE as a pdf file. Please also submit the Appendix file at a higher resolution. The images are currently pixelated and not publishable at the current resolution.
- Materials and Methods should be just Methods.
- MS title should be removed from page 2.
- Abbreviations section needs to be removed from the manuscript. Abbreviations should be defined in brackets after their first mention in the text, not in a list of abbreviations.

Figure Legends - Comments

- Please define the annotated p values ****/****/**/* as well as provide the exact p-values for the same in the legend of figure 1D, E, G, H; 2B, C, D, E, F, H; 3G, I, K; 5F, H; 6F-H; EV1 D, E; EV2 B, C; EV3 D, E; EV4 C, F; EV5 B, C, G, H, K as appropriate and reasonable.
- Please indicate the statistical test used for data analysis in the legends of figures 1D, E, G, H; 2B, C, D, E, F, H; 3G, I, K; 5F, H; 6F-H; EV1 D, E; EV2 B, C; EV3 D, E; EV4 C, F; EV5 B, C, G, H, K
- Please note that the arrow heads are not defined in the legend of figure 1F. This needs to be rectified.
- Please note that the dotted borders are not defined in the legend of figure 1B. This needs to be rectified.

I would like to suggest a few minor changes to the abstract that needs to be written in present tense. Please let me know whether you agree with this:

Mitochondrial DNA (mtDNA) serves as a potent activator for cellular innate immune responses. Topoisomerase 3 α (TOP3 α), a type IA topoisomerase, is canonically localized to mitochondria and nuclei, but its enigmatic cytosolic fraction-observed over two decades ago-has remained functionally undefined. Here, we uncover a critical role for cytosolic TOP3 α in amplifying mtDNA-triggered innate immunity. We observe that aberrant TOP3 α expression causes mtDNA clustering and release via mPTP-VDAC, stimulating cGAS-mediated inflammatory responses. Cytosolic TOP3 α facilitates the sensing of released mtDNA by cGAS and amplifies downstream innate immune signaling. Using an in vitro cell-free system, we reveal that TOP3 α directly augments mtDNA interaction with cGAS, which in turn competes with TOP3 α for mtDNA binding. A rare mutation of a highly conserved residue (G250D) of TOP3 α impairs the assembly of TOP3 α polypeptides into protein complexes and its binding to mtDNA. Furthermore, mutant TOP3 α hinders cGAS-mtDNA interaction and compromises cGAS-driven immunity. Our findings reveal a function for cytosolic TOP3 α as a regulator for cGAS-driven inflammation.

EMBO press papers are accompanied online by A) a short (1-2 sentences) summary of the findings and their significance, B) 2-3 bullet points highlighting key results and C) a synopsis image that is exactly 550 pixels wide and 200-600 pixels high (the height is variable). The synopsis image should provide a sketch of the major findings, like a graphical abstract. Please note that text needs to be readable at the final size. Please send us this information along with the final manuscript.

I look forward to seeing a final version of your manuscript as soon as possible.:

Best regards,

Esther

Referee #1:

The authors have improved the manuscript and addressed most concerns raised. One thing to clarify is whether the qPCR data is just reporting 3 technical replicates i.e they've only done one experiment (with three technical repeats within it) for all their qPCR. If that is the case, this is insufficient.

Referee #2:

The authors have resolved my questions with reasonable experiments and explanations. I recommend its publication now.

Referee #3:

I thank the authors for their thorough revisions. All of my concerns have been adequately addressed, I have no further comments.

Editorial requests:

Thank you for the submission of your revised manuscript. We have now received the enclosed reports from the referees, and I am happy to say that all support its publication now. Please address or comment on referee 1's concern. Providing exclusively technical replicates for qPCR is indeed not sufficient.

Response: Thank you for your kind help during the peer review process. We agree with you and the reviewer 1 that only 3 technical replicates in one experiment are not sufficient. We have included the data from independent repeat experiments in the Appendix Figures for reference.

A few editorial requests will also need to be addressed before we can proceed with the official acceptance of your manuscript.

- Your ms does not contain a "Data Availability Section" (DAS) that needs to be placed before the Acknowledgments. If you have not generated data in this study that is deposited in public databases please mention this fact in the DAS.

Response: Thank you for your note. We have added the "Data Availability Section" as requested.

- The conflict of interest subheading needs to be renamed to "Disclosure and Competing Interests Statement"

Response: We have revised the relevant section as requested.

- The REFERENCE format is not correct: et al needs to be used after 10 author names. Please use the EMBO reports reference style.

Response: We have revised the relevant section as requested according to the Author Guidelines-References of EMBO reports.

- The FUNDING INFO is OK, but it needs to go under Acknowledgments and the "Funding" section heading is not needed.

Response: As suggested, we have incorporated the funding information into the Acknowledgments section.

- Please upload the APPENDIX FILE as a pdf file. Please also submit the Appendix file at a higher resolution. The images are currently pixelated and not publishable at the current resolution.

Response: We have re-generated the high-resolution images and compiled them into a single PDF file.

- Materials and Methods should be just Methods

Response: We have revised the relevant section as requested.

- MS title should be removed from page 2.

Response: We have revised the relevant section as requested.

- Abbreviations section needs to be removed from the manuscript. Abbreviations should be defined in brackets after their first mention in the text, not in a list of abbreviations.

Response: We have revised the manuscript as requested.

Figure Legends - Comments

- Please define the annotated p values ****/**/**/* as well as provide the exact p-values for the same in the legend of figure 1D, E, G, H; 2B, C, D, E, F, H; 3G, I, K; 5F, H; 6F-H; EV1 D, E; EV2 B, C; EV3 D, E; EV4 C, F; EV5 B, C, G, H, K as appropriate and reasonable.

Response: Thank you for your kind request. We find it hard to include all exact p-values in the figure legend which makes the legend hard to read. Thus, we have now specified the exact p-values in the figures as requested.

- Please indicate the statistical test used for data analysis in the legends of figures 1D, E, G, H; 2B, C, D, E, F, H; 3G, I, K; 5F, H; 6F-H; EV1 D, E; EV2 B, C; EV3 D, E; EV4 C, F; EV5 B, C, G, H, K

Response: We have now specified the statistical methods used in the figure legends.

- Please note that the arrow heads are not defined in the legend of figure 1F. This needs to be rectified.

Response: Thank you for your kind request. We have now clarified the meaning of the arrow heads in the figure legend.

- Please note that the dotted borders are not defined in the legend of figure 1B. This needs to be rectified.

Response: Thank you for your kind request. We have now clarified the meaning of the dotted borders in the figure legend.

- I would like to suggest a few minor changes to the abstract that needs to be written in present tense. Please let me know whether you agree with this:

Mitochondrial DNA (mtDNA) serves as a potent activator for cellular innate immune responses. Topoisomerase 3 α (TOP3 α), a type IA topoisomerase, is canonically localized to mitochondria and nuclei, but its enigmatic cytosolic fraction-observed over

two decades ago-has remained functionally undefined. Here, we uncover a critical role for cytosolic TOP3 α in amplifying mtDNA-triggered innate immunity. We observe that aberrant TOP3 α expression causes mtDNA clustering and release via mPTP-VDAC, stimulating cGAS-mediated inflammatory responses. Cytosolic TOP3 α facilitates the sensing of released mtDNA by cGAS and amplifies downstream innate immune signaling. Using an in vitro cell-free system, we reveal that TOP3 α directly augments mtDNA interaction with cGAS, which in turn competes with TOP3 α for mtDNA binding. A rare mutation of a highly conserved residue (G250D) of TOP3 α impairs the assembly of TOP3 α polypeptides into protein complexes and its binding to mtDNA. Furthermore, mutant TOP3 α hinders cGAS-mtDNA interaction and compromises cGAS-driven immunity. Our findings reveal a function for cytosolic TOP3 α as a regulator for cGAS-driven inflammation.

Response: Thank you for your kind suggestion. We have revised the abstract accordingly.

- EMBO press papers are accompanied online by A) a short (1-2 sentences) summary of the findings and their significance, B) 2-3 bullet points highlighting key results and C) a synopsis image that is exactly 550 pixels wide and 200-600 pixels high (the height is variable). The synopsis image should provide a sketch of the major findings, like a graphical abstract. Please note that text needs to be readable at the final size. Please send us this information along with the final manuscript.

Response: We have now added a short summary, 3 bullet points and a synopsis image as requested.

Referee #1:

The authors have improved the manuscript and addressed most concerns raised. One thing to clarify is whether the qPCR data is just reporting 3 technical replicates i.e they've only done one experiment (with three technical repeats within it) for all their qPCR. If that is the case, this is insufficient.

Response: We agree with you and the editor that only 3 technical replicates in one experiment are not sufficient. We have included the data from independent repeat experiments in the Appendix Figures for reference. Thank you for your kind help during the peer review process.

Referee #2:

The authors have resolved my questions with reasonable experiments and explanations. I recommend its publication now.

Response: Thank you for your kind help during the peer review process.

Referee #3:

I thank the authors for their thorough revisions. All of my concerns have been adequately addressed, I have no further comments.

Response: Thank you for your kind help during the peer review process.

Prof. Jian Qiu
Xiangya Hospital, Central South University
Hunan Key Laboratory of Molecular Precision Medicine
Xiangya Road 87
Changsha, Hunan 410008
China

Dear Prof. Qiu,

I am very pleased to accept your manuscript for publication in the next available issue of EMBO reports. Thank you for your contribution to our journal.
